# TIME CONDITIONED FORESEEING: TEMPORAL GENERATIVE PRETRAINING FOR EHR FOUNDATION MODELS

## ABSTRACT

Electronic Health Records (EHRs) possess unique characteristics that differ significantly from natural language. However, existing models have overlooked these properties and largely relied on Natural Language Processing (NLP) approaches, resulting in suboptimal performance. To address these limitations, we propose a pretraining method designed to effectively capture the distinctive features of EHRs. First, EHRs contain both clinically critical and less informative numerical ranges. To reflect this, we introduce a Pathology-Focused Binning strategy that emphasizes values with clinical significance. Second, both absolute timestamps and relative time intervals are important in EHRs. To incorporate these temporal aspects, we propose a Dual-Calendar Rotary Positional Embedding (RoPE) that jointly encodes complementary temporal signals. Third, many medical applications require modeling long-term patient interactions. Accordingly, we extend conventional next-token prediction with a Time-Conditioned Foreseeing (TCF) objective, enabling the model to forecast long-range clinical events across multiple temporal horizons. Our approach establishes the first genuine temporal generative EHR model, advancing long-range clinical forecasting. It outperforms existing EHR foundation models on seven diverse downstream tasks and enables realistic and temporally consistent EHR generation. All code and models will be made publicly available in the final version of the manuscript.

## 1 INTRODUCTION

Electronic health records (EHRs) are longitudinal records that comprehensively document a patient's medical history. EHRs help clinicians assess patient conditions, coordinate diagnostic and therapeutic interventions, and communicate with other healthcare providers (Häyrinen et al., 2008). One of the key objectives in medical AI is to develop models that can learn from EHRs to perform various clinical tasks. However, building such models is challenging due to the complex temporal dependencies and the predominance of numerical data in EHRs (Nasarudin et al., 2024). Recently, there have been growing efforts to leverage large language model (LLM) training paradigms in building EHR foundation models (Niu et al., 2024). Despite these advances, approaches explicitly designed to model the distinct characteristics of EHRs are still in their early stages of development.

EHRs consist of diverse clinical events—such as examinations, treatments, and diagnoses—that are recorded with associated timestamps. Figure 1 illustrates an example EHR, where events are arranged chronologically, and shows how these events can be transformed into a sentence of tokens. Recent preprocessing approaches for EHRs commonly represent a single clinical event as a `Time` (T), `Feature` (F), `Value` (V) triplet (Tipirneni & Reddy, 2022). Here, the `Feature` denotes attributes such as diagnosis codes, prescribed medications, or laboratory tests (e.g., Systolic Blood Pressure) and represented as a single token, while the `Value` corresponds to the result or auxiliary information of the `Feature` (e.g., 87mmHg). `Values` are typically numerical but may also be absent, or take the form of heterogeneous modalities such as text, depending on the `Feature`. Despite the necessity of including all triplet components for a faithful representation of clinical events, as indicated in the "data usage" column of Table 1, even the most recent EHR foundation models often exclude `Time` or `Value` information due to modeling complexities (Yang et al., 2024).

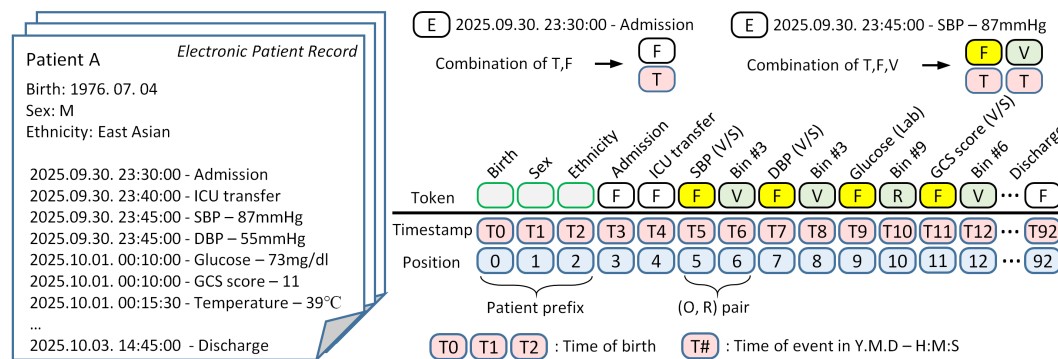

Figure 1: (Left) Extraction of raw patient data from the EHR database in chronological order. (Right) Tokenization of each event (E) with triplet representation, where patient information is placed at the beginning, `Features` and `Values` are tokenized, and timestamps remain continuous.

Recent EHR foundation models have improved performance on various downstream tasks through large-scale pre-training. However, most of these models follow standard LLM training paradigms without adapting to the structure and clinical semantics of EHR data (Burkhart et al., 2025), which differ from natural language. For example, converting temporal information into absolute positional embeddings hinders capturing relative intervals and preserving clinically meaningful calendrical information (Likhomanenko et al., 2021). Also, processing numeric `Value` through *uniform* binning concentrates bins around normal ranges and reduces resolution for pathological states. Moreover, most learning objectives are adopted from language modeling, such as next-token prediction (NTP) or masked language modeling (MLM), without considering EHR-specific characteristics. To address these limitations, we introduce improved binning, temporal embedding, and novel training objectives tailored to EHR data and clinical planning process.

First, we introduce a simple yet effective **Pathology-focused Binning** for `Value` tokenization. As shown in the "Value Binning" column of Table 1, most EHR models tokenize `Value` through uniform binning. However, as illustrated in Figure 2A, uniform binning assigns a large amount of bins to physiologic ranges, while allocating only a few bins to clinically important pathologic ranges, thereby limiting the ability to distinguish the severity of abnormalities. Other models rely on false distributional assumptions of Gaussianity, and instead apply standard deviation (std)–based binning (Zhu et al., 2024) or z-normalization (Tipirneni & Reddy, 2022), making them vulnerable to outliers, long-tailed, and dual peaks distributions common in EHR. To address this, we propose a density-based binning that makes no distributional assumptions and focuses on pathological ranges. In this approach, values in high-density physiologic zones are assigned lower weights, whereas values in low-density pathologic zones receive higher weights. This design is suited for all value distributions, and we are the first to apply such binning to EHR models.

Second, we introduce **Dual-Calendar RoPE**, a novel timestamp addressing method for EHRs. Unlike language models, where tokens are assumed to be uniformly spaced, EHRs contain events with highly irregular intervals. Clinically, both relative intervals and calendarical context—e.g., morning/afternoon or weekday/weekend—are important (body temperature is higher in the afternoon, and dialysis complications are common after weekends (Fotheringham et al., 2020)). Also, multiple events may occur at the same time, such as laboratory tests recorded together. As shown in Figure 2B, we partition the dimensions of rotary positional embedding (Su et al., 2024) to jointly encode position and time, assigning calendrical components (e.g., minute, day, month) in increasing units to the time dimension. This enables explicit modeling of distance relations such as "two tests performed at the same time"[1] or "the same test performed at the same hour on different days." The "Time Addressing" column of Table 1 shows that conventional models have not fully addressed crucial temporal information.

Finally, and most importantly, we propose a new learning objective, **Time-Conditioned Foreseeing** (TCF). This objective aligns with the clinical process of treatment planning, and it enables, for the

---

[1]Suppose that SBP, 120mmHg, DBP, 80mmHg are recorded simultaneously. The position dimension provides additional support to prevent the model from confusing results such as SBP, 80mmHg, DBP, 120mmHg.

Table 1: Comparison of recent EHR models by architecture and training objective. **Data usage**: whether the model uses `Time` and `Value` information. Value binning: whether values are uniformly binned (STraTS embeds values continuously) and whether bin tokens are shared across `Features`. **Time addressing**: whether the model considers relative time intervals, calendrical time, and distinguishes concurrent events; same timestamps. **Learning objective**: type of loss, **'Foresee'** - whether model forecasts beyond the next-token, and **'Temporal Generation'** - whether temporal generative modeling is possible. Abbreviations*: MEP; missing entity prediction, TTE; time to event prediction, TCF; time conditioned foreseeing. Refer to the related works section for details[†].

| Models | [Data usage] | | [Value Binning] | | [Time addressing] | | | [Learning Objective] | | |
|---|---|---|---|---|---|---|---|---|---|---|
| | Event Timestamp | Numeric Value | Non-uniform | Value sharing | Relative interval | Calendrical time | Non-concurrency | Type | Foresee | Temporal Generation |
| BEHRT (2020) Med-BERT (2021) Foresight (2024) ClinicalMamba (2024) EHR-BERT (2024) | X | X | - | - | X | X | - | NTP&MLM | X | X |
| HEART (2024) FM4EHR (2025) | X | O | X | X O | X | X | - | MEP* NTP | X | X |
| MOTOR (2024) | O | X | - | | O | X | X | TTE* | O | X |
| STraTS (2022) | O | O | O | O | X | X | - | MSE | X | X |
| EHRSHOT (2023) | O | O | X | X | X | X | - | NTP | X | X |
| TRADE (2024) | O | O | O | X | X | X | - | MLM | X | X |
| EHRMamba (2025) | O | O | X | X | X | X | - | NTP&MLM | X | X |
| ETHOS (2024) | O | O | X | O | △[†] | X | - | NTP | X | △[†] |
| OURS | O | O | O | both | O | O | O | TCF* | O | O |

first time, generative temporal modeling of a patient's medical timeline. As shown in the "Learning Objective" column of Table 1, prior models have relied on objectives designed for language models or variants thereof, with the exception of the time-to-event (TTE) objective. Conventional EHR models trained with NTP loss capture only $P(F_{next} \mid E_{past})$, without explicitly modeling temporal information. Consequently, they cannot distinguish whether an event occurs minutes later or after many hours, treating both urgent and routine vital sign measurements (short and long time intervals respectively) identically as the 'next token.'

In contrast, TCF explicitly models long-range temporal information, thereby capturing how real-world clinical practice unfolds over time. In NLP, missing a single token disrupts grammar, and consecutive tokens are tightly correlated. By contrast, neighboring EHR events are loosely connected and often exhibit long-range dependencies, such as 8-hour follow-up tests. This reflects clinical practice, where physicians do not always act in real time but instead devise broader clinical plans. TCF embodies this principle: rather than the short-sighted scope of NTP, which predicts only the immediate next event, TCF enables questions such as, "What intervention is needed in the next six hours?" To achieve this, TCF module first generates the next timestamp from the last hidden state. The multiple foreseeing timestamps are then fed back as module inputs, conditioning subsequent token generation. This time-conditioned architecture allows simultaneous learning of $P(T_{next} \mid E_{past})$ and $P(F_{foresees} \mid T_{foresees}, E_{past})$, leading to improved performance.

Our model ranked first across all combinations of the three dataset configurations and seven diverse downstream tasks. Across these tasks, the AUPRC was consistently improved, reaching up to 48% higher than that of the second-best model, highlighting a clinically meaningful improvement given the data imbalance. We also demonstrated that the model generates temporally stable, realistic EHR records and is capable of leveraging the calendrical component in generative modeling.

Our contributions can be summarized as follows:

- Pathology-Focused binning: Introduces density-adjusted binning to the EHR foundation model, focusing on clinically relevant pathologic ranges.

- Dual-Calendar RoPE: Simultaneously represents both calendrical time and positional information, allowing model to capture calendrical periodicity and event concurrency.

- Time Conditioned Foresee Objective: Enables clinically aligned foreseeing training and temporal generative modeling of patient medical timelines.

## 2 RELATED WORKS

EHR foundation models differ from medical specialist LLMs, which rely on patient history texts summarized by clinicians. EHR foundation models learn directly from raw EHR events (Burkhart et al., 2025) and have been applied to various downstream clinical tasks (Table 1).

A common practice in EHR modeling is to represent each EHR event as a triplet of `Time`, `Feature`, and `Value` (Tipirneni & Reddy, 2022; Lee et al., 2023). However, many models exclude temporal and numeric data, as they are difficult to handle in standard language model frameworks. For instance, BEHRT (Li et al., 2020), Med-BERT (Rasmy et al., 2021), and others rely solely on discrete `Features`, omitting critical information and limiting their utility.

Some models incorporate numeric `Values` but omit `Time`. HEART (Huang et al., 2024) discretize `Values` into uniform bins, mapping `Feature–Value` pairs to single tokens. This approach inflates the vocabulary size, leading to data sparsity. FM4EHR (Burkhart et al., 2025) addresses this by tokenizing `Features` and `Values` separately, allowing tokens to be shared.

In contrast, MOTOR (Steinberg et al., 2024) models `Time` but not `Value`, performing survival analysis by treating each feature's occurrence as an endpoint. Its utility is limited by its inability to handle numeric values, low temporal expressiveness based on pre-defined intervals, unrealistic constant hazard assumption, and a quadratic complexity that hinders practical application. Moreover, encoding timestamp as 'days since birth' with RoPE does not account for calendrical time.

STraTS (Tipirneni & Reddy, 2022) tokenizes only the `Feature`, embedding `Value` and `Time` as continuous variables to predict the next value. By modeling only $P(V_{next} \mid E_{past})$, it loses important context and cannot support generative modeling.

TRADE (Zhu et al., 2024) and EHRmamba (Fallahpour et al., 2025) used MLM/NTP paradigms, discretizing values and applying absolute positional embeddings to `Feature` and `Value` tokens.

ETHOS (Renc et al., 2024) tokenizes time intervals and insert time-interval tokens between events. This coarse discretization limits medical precision, cause cumulative errors, and increases computational cost by lengthening the sequence. Unlike positional embeddings, it requires aggregating all intervening tokens to determine a time duration. More details are provided in Appendix A

To address these limitations, this work designs modeling strategies and learning objectives tailored to the unique characteristics of EHR data.

## 3 METHOD

**Pathology-Focused Binning**. First, we estimate the value distribution non-parametrically using a Gaussian Kernel Density Estimator (KDE). $V_{list}^f$ denotes the list of all `Values` of `Feature` $f$ in the training set. We uniformly partition the value range $[\min(V_{list}^f), \max(V_{list}^f)]$ with $X = \{x_1, x_2, \ldots, x_P\}$, where the inverval is $0.05\sigma$. At each discrete point $x_k \in X$, data density $\rho(x_k)$ is calcuated with Gaussian convolution kernel from all value $v_j \in V_{list}^f$. The density is:

$$\rho(x_k) = \sum_{j=1}^{|V_{list}^f|} K_h(x_k - v_j), \quad s.t. \ K_h(u) = \exp\left(-\frac{u^2}{2(0.1\sigma)^2}\right)$$

This allows us to approximate the local density $\rho(v)$ for any given value $v$. Then, we assign a weight $w(v)$ to each value that is inversely proportional to its density, effectively giving greater importance to values in sparser region ($w(v) \propto \rho(v)^{-N}; N \geq 1$). In short, values in sparse regions are assigned larger weights than those in dense regions.

Second, these density-based weights are used to construct the final value bins via weighted percentile binning. In this step, the contribution of each unique value $v_j$ with a raw count of $c_j$ is scaled by its weight $w(v_j)$, creating a weighted count $c_j' := c_j \cdot w(v_j)$.

Bin thresholds are then determined from the cumulative distribution of these weighted counts. As a result, high-weight values from pathologic ranges command a larger share of the percentile space, leading to a finer-grained partitioning in these clinically important areas (Figure 2A). The detailed methodology is described in Appendix B.1.

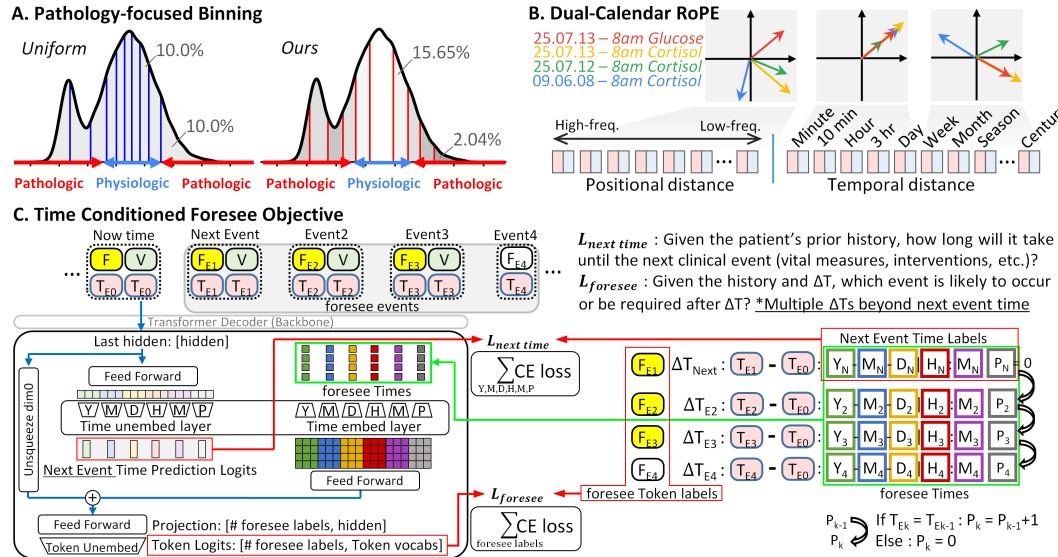

Figure 2: (A) Uniform binning concentrates bins in dense, physiologic ranges. In contrast, our density-based method allocates more bins to medically significant pathologic ranges. (B) Events at the same time are distinguished by their positional distance. Events occurring at the same time on different dates share the same representation for time units below a day but have different representations for units of a day or longer. (C) Illustrates TCF objective of a single timestep (actual model training is fully parallel, like NTP). The TCF objective consists of $L_{next\ time}$ and $L_{foresee}$. The last hidden state is passed through a time head to predict the interval to the next event in a calendrical format ($L_{next\ time}$). Then, the times to multiple future events are re-input and combined with the last hidden state to predict the events at those specific times ($L_{foresee}$).

**Dual-Calendar Rotary Position Embedding**. Second, we propose a novel positional encoding designed for the temporal characteristics of EHR (Figure 2). It jointly models the relative order and calendrical interval by partitioning the dimension of each query and key vector, $x \in \mathbb{R}^d$, into a positional component $x_{pos} \in \mathbb{R}^{d_{pos}}$ and a temporal component $x_{time} \in \mathbb{R}^{d_{time}}$ ($d = d_{pos} + d_{time}$):

$$x = [x_{pos} \| x_{time}]$$

The $x_{pos}$ component uses a standard RoPE to encode the relative token position, $p$. With a reduced dimensionality ($d \rightarrow d_{pos}$), it employs a truncated frequency spectrum. This strategic choice focuses its role on disambiguating the order of co-occurring events sharing an identical timestamp, while long-range dependencies are handled by the temporal component. The rotation angle is defined as:

$$\theta_{p,i}^{(pos)} = \frac{p}{10000^{2i/d}} \ , \ i \in \{0, 1, ..., d_{pos}/2 - 1\}$$

The core of our method, the $x_{time}$ component, encodes the second-level timestamp $t$. This is achieved using a predefined set of semantically meaningful calendrical periods (e.g., minute=60s, hour=3600s,...; see Table 5 for a full list). For each period $s_j$ in the set, a rotation angle $\theta_{t,j}^{(time)}$ is calculated as the phase of the event within that period:

$$\theta_{t,j}^{(time)} = \left( \frac{t \pmod{s_j}}{s_j} \right) \cdot 2\pi \ , \ j \in \{0, 1, ..., d_{time}/2 - 1\},$$

The two components are rotated independently using their respective angles and then concatenated to form the final query vector $q'$ (and also for the key). This allows the attention mechanism to simultaneously address both sequential order and calendrical time (More details in Appendix B.2).

$$q' = [\text{RoPE}(q_{pos}, \theta^{(pos)}) \| \text{RoPE}(q_{time}, \theta^{(time)})]$$

**Time-Conditioned Foresee Objective (TCF)**. Lastly, we propose a novel learning objective to effectively model the temporal dynamics of EHR data. TCF employs a dual-objective structure (Figure 2C) that simultaneously learns to: (1) predict *when* the next event will occur ($P(\Delta T_{\text{next}}|E_{\text{past}})$),

and (2) foresee *what* event will happen at a specified future time ($P(F_{\text{foresee}}|\Delta T_{\text{foresee}}, E_{\text{past}})$), unlike NTP which only models $P(F_{\text{next}}|E_{\text{past}})$.

The TCF module is placed after the transformer backbone. It takes the final hidden state $h_{\text{last}} \in \mathbb{R}^{d_{\text{model}}}$ as input[2] and outputs both a next time prediction loss and a conditioned hidden states for future event prediction.

To generate a calendrical ground-truth label for $\Delta T_{\text{next}}$, the time delta, expressed in seconds, is transformed into an integer vector of dimension $N_{\text{scales}}$. Each element of this vector corresponds to a predefined calendrical time unit, ranging from `10-year` to `1-minute` (e.g., $[\alpha, \beta, \gamma, \dots]$ represents a time composed of $\alpha$ years, $\beta$ months, $\gamma$ days, etc.).

To predict $\Delta T_{\text{next}}$ from $h_{\text{last}}$, the last hidden is projected into #$N_{\text{scales}}$ vectors of size $d_{\text{embed}}$. Each of these vectors is transformed into time-logit through the unembedding layer.

$$h_{\text{time}} = \text{FFN}_{\text{enc}}(h_{\text{last}}) \in \mathbb{R}^{(N_{\text{scales}} \cdot d_{\text{embed}})} \rightarrow \{h_{\text{time}}^{(i)}\}_{i=1}^{N_{\text{scales}}} , \quad \text{time-logits}^{(i)} = h_{\text{time}}^{(i)} \cdot (W_{\text{embed}}^{(i)})^T$$

$\mathcal{L}_{\text{next\_time}}$ is Cross-Entropy loss between these $\{\text{time-logit}_{\text{time}}^{(i)}\}_{i=1}^{N_{\text{scales}}}$ and the calendrical $\Delta T_{\text{next}}$ labels, averaged over $N_{\text{scales}}$.

For foreseeing future events, a **Time-Conditioning** process is performed. We aim to predict the `Feature` of $N_{\text{foresee}}$ future events. A given future time deltas, $\Delta T_{\text{foresee}}$, is first transformed into a vector of integer labels ($C_{\text{foresee}} \in \mathbb{Z}^{N_{\text{foresee}} \times N_{\text{scales}}}$) using the same multi-scale decomposition. These labels are passed through embedding layers to produce a comprehensive time embedding, $e_{\text{time}} \in \mathbb{R}^{N_{\text{foresee}} \times (N_{\text{scales}} \cdot d_{embed})}$. Finally, this time embedding is fused with the original hidden state $h_{\text{last}}$ via a residual connection to produce a time conditioned hidden state, $h_{\text{conditioned}}$.

$$h_{\text{conditioned}} = \text{FFN}(\text{LayerNorm}(h_{\text{last}} + \text{FFN}(e_{\text{time}}))) \in \mathbb{R}^{N_{\text{foresee}} \times d_{\text{model}}}$$

This $h_{\text{conditioned}}$ is projected to token-logit $\in \mathbb{R}^{N_{\text{foresee}} \times \text{vocab\_size}}$ that predicts the clinical event (`Feature`) that occur at the corresponding future timestamps.

$\mathcal{L}_{\text{foresee}}$ is Cross-Entropy loss between the future events and the token-logit, averaged over $N_{\text{foresee}}$. Through this dual-objective learning ($\mathcal{L} = \mathcal{L}_{\text{next\_time}} + \mathcal{L}_{\text{foresee}}$), our model acquires the ability to accurately and generatively model a patient's entire medical timeline.

So far, we have considered the position where `Feature` is predicted given the previous events. Modeling `Value` given the previous events and `Feature` is carried out in the same manner. Since F and V belong to the same event and thus share the time label, we always have $\Delta T_{\text{next}} = 0$. Moreover, because $V$ is conditioned on the preceding $F$, we predict $V_{\text{now}}$ by modeling

$$P(V_{\text{foresee}} \mid \Delta T_{\text{foresee}}, F_{\text{now}}, E_{\text{past}})$$

while inserting only a zero into $\Delta T_{\text{foresee}}$. More detailed explanation and tensor-level parallel processing are provided in Appendix B.3.

## 3.1 DATA AND PREPROCESSING

While many EHR models rely on private datasets and often do not release their code or parameters—making reproduction and evaluation difficult—we use a publicly available dataset and provide open-source code throughout all stages. Specifically, we employ the MIMIC-III Clinical Database v1.4 (Johnson et al., 2016a), which contains comprehensive clinical data from over 30,000 patients. We adopt the widely used preprocessing and train/test split pipeline introduced by Harutyunyan et al. (2019). A summary of the dataset is provided in Table 2. Further details are provided in Appendix C. Tasks necessitating clinical judgment, such as defining exclusion criteria and outlier removal, were independently reviewed by an internist, an otolaryngologist, and a general physician.

Table 2: Data summary. Parentheses indicate cases where bins are not shared.

| MIMIC-III preprocessed | Train / Test |
| --- | --- |
| Total Patient # | 28,728 / 5,070 |
| Total Hospitalization # | 35,730 / 6,295 |
| Total Events # | 38,641,175 / 6,744,906 |
| Total Tokens # | 77,109,833 / 13,459,430 |
| Avg. length | 2,684 / 2,655 |
| Max length | 393,337 / 62,759 |
| Unique Tokens # | 155 (1,208) |
| Token # bin | 10 |
| Token # ethnicity | 10 |
| Token # vital signs | 17 |
| Token # laboratory tests | 100 |

---

[2]In practice, the full last hidden state $H_{\text{last}} \in \mathbb{R}^{B \times L \times d_{\text{model}}}$ is processed in parallel, similar to the NTP.

## 3.2 Backbone architecture, Baseline models, and Pre-training

We used a Transformer decoder as the backbone for all experiments. For Dual-calendar RoPE, the first 24 dimensions of the 64-dim K and Q vectors encode positional information, and the remaining 40 dimensions encode calendric time. Baseline models were reproduced under identical conditions, including backbone and training data. We mostly followed the original papers' implementations but made necessary modifications where direct application was infeasible (e.g., adapting the MOTOR model to numeric value events). The pre-training input token length was fixed at 2048. Sequences exceeding this length (Appendix Figure 7) were segmented with a 512-token overlap, and we ensured that a single event's F and V were not split at the segmentation point. A detailed description of our model, baselines, and pre-training can be found in Appendix D.

# 4 Result

## 4.1 Downstream task and Fine-tuning

Table 3: Results on downstream tasks using EHR datasets with 117, 17, and 6 features. The Test loss column reports the overall test loss for each feature set (lower is better). For 117 features, we report performance on seven downstream tasks ranging from IHM to Vaso. Binary classification tasks are measured by AUROC (ROC) and AUPRC (PRC), while multiclass tasks are evaluated with macro F1 (Ma-f1) and Cohen's Kappa. For tasks with multiple subtasks, both macro and micro AUROC are reported. We trained our model with and without value sharing; in both cases, it outperformed all other baselines. Full downstream task results are provided in Appendix Table 9-11)

| Tasks | Test Loss (↓) | | | IHM | | Phe | | Dec-death | | Dec-arrest | | LOS | | HUO | | Vaso | |
|---|---|---|---|---|---|---|---|---|---|---|---|---|---|---|---|---|---|
| Metric | 117 | 17 | 6 | ROC | PRC | macro | micro | ROC | PRC | ROC | PRC | Ma-f1 | Kappa | macro | micro | ROC | PRC |
| **No Value share** | | | | | | | | | | | | | | | | | |
| HEART | 5.304 | 5.434 | 5.835 | 0.838 | 0.442 | 0.717 | 0.718 | 0.869 | 0.205 | 0.862 | 0.199 | 0.150 | 0.142 | 0.703 | 0.701 | 0.865 | 0.363 |
| MOTOR | 4.945 | 5.212 | 5.645 | 0.872 | 0.547 | 0.770 | 0.773 | 0.904 | 0.272 | 0.889 | 0.261 | 0.174 | 0.163 | 0.753 | 0.748 | 0.891 | 0.438 |
| EHRSHOT | 5.841 | 6.078 | 6.341 | 0.801 | 0.433 | 0.634 | 0.633 | 0.829 | 0.167 | 0.802 | 0.153 | 0.101 | 0.115 | 0.701 | 0.614 | 0.867 | 0.341 |
| TRADE | 5.260 | 5.454 | 6.048 | 0.828 | 0.441 | 0.738 | 0.738 | 0.867 | 0.170 | 0.857 | 0.165 | 0.158 | 0.151 | 0.732 | 0.731 | 0.869 | 0.393 |
| EHRmamba | 5.137 | 5.439 | 5.926 | 0.868 | 0.557 | 0.690 | 0.687 | 0.901 | 0.277 | 0.886 | 0.260 | 0.150 | 0.159 | 0.751 | 0.753 | 0.873 | 0.399 |
| Ours (No share) | **4.686** | **4.907** | **5.367** | **0.889** | **0.607** | **0.809** | **0.816** | **0.928** | **0.400** | **0.917** | **0.388** | **0.181** | **0.185** | **0.776** | **0.781** | **0.912** | **0.498** |
| **Value share** | | | | | | | | | | | | | | | | | |
| FM4EHR | 6.429 | 6.389 | 6.397 | 0.617 | 0.177 | 0.530 | 0.519 | 0.744 | 0.075 | 0.778 | 0.102 | 0.023 | 0.003 | 0.598 | 0.653 | 0.690 | 0.130 |
| ETHOS | 4.971 | 5.248 | 5.572 | 0.859 | 0.530 | 0.739 | 0.746 | 0.900 | 0.311 | 0.890 | 0.304 | 0.170 | 0.165 | 0.721 | 0.731 | 0.890 | 0.437 |
| STraTS | 5.786 | 5.812 | 6.071 | 0.759 | 0.311 | 0.656 | 0.661 | 0.840 | 0.141 | 0.804 | 0.103 | 0.123 | 0.121 | 0.590 | 0.598 | 0.864 | 0.331 |
| Ours (Share) | **4.879** | **5.043** | **5.561** | **0.876** | **0.559** | **0.781** | **0.784** | **0.910** | **0.319** | **0.902** | **0.310** | **0.173** | **0.170** | **0.749** | **0.755** | **0.906** | **0.470** |

We evaluated our model on a range of clinical downstream tasks commonly used in EHR model evaluation. These tasks, defined by clinical labels excluded from training, are not direct measures of generative modeling performance but serve as proxies for the quality of patient representations. In addition to the four MIMIC-III benchmark (Harutyunyan et al., 2019) tasks—In-hospital Mortality (IHM), Decompensation-death (Dec-death), Length of Stay (LOS), and Phenotyping (Phe)—we included three additional tasks: Decompensation-arrest (Dec-arrest), Oliguria/Anuria (HUO), and Vasopressor (Vaso) use. Label counts for all tasks are provided in Appendix Table 8, with detailed descriptions in Appendix E.1.

Table 4: Ablation study on Pathology-Focused Binning (Binning), Dual-Calendar Rotary Positional Embedding (Embedding), and Time-Conditioned Foreseeing (Objective).

| Binning | Embedding | Objective | Test loss |
|---|---|---|---|
| V | V | V | 4.686 |
| Uniform | V | V | 4.713 |
| Uniform | RoPE | V | 4.810 |
| Uniform | RoPE | NTP | 5.241 |

Downstream task-specific prediction heads were attached to the backbone. Since labels must be inferred using only information up to each timestep, a causal mask was applied for all baselines. To evaluate generalization to data with different distributions (e.g., missing lab information), we experimented with three input configurations: all 117 features, 17 vital signs (without lab data), and only 6 vital signs (SBP, DBP, body temperature, heart rate, respiratory rate, SpO2). Please refer to Appendix E.2 for more details.

Table 3 summarizes the results on downstream tasks. To ensure fair comparison, we trained our model with and without value sharing and compared each setting to the corresponding baselines. In

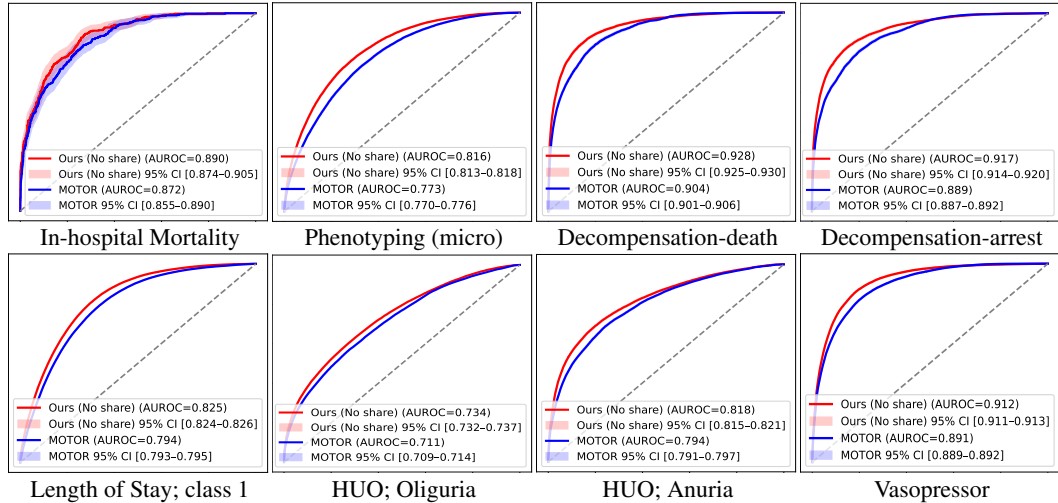

Figure 3: AUROC curves of our model and the second-best baseline, with 95% confidence intervals estimated via bootstrapping. LOS was evaluated as a binary classification for the first class, and Phenotyping was assessed using micro-ROC. For HUO, both oliguria and anuria are presented.

both cases, our model consistently outperformed all baselines across the three input configurations and all downstream tasks. Notably, for the decompensation task, which predicts patient death or arrest up to 24 hours in advance, our model achieved an AUPRC nearly 50% higher than that of the second-best model. Given the severe class imbalance in these tasks (positive:negative ratio of 1:40), this represents a significant improvement in real-world clinical settings where high precision is crucial. Additionally, the ablation study (Table 4) shows that all three of our proposed methods contribute substantially to the performance improvement.

Figure 3 shows the ROC curves with 95% confidence intervals, confirming that our model achieves statistically significant improvements over the second-best model in most tasks. The complete results for all three input configurations can be found in Appendix F.1.

### 4.2 TEMPORAL GENERATIVE MODELING

Our model is the first to generate fine-grained temporal information and clinical events conditioned on time, demonstrating strong temporal generative modeling of EHR data. To qualitatively assess its effectiveness, we compared it (share ver. for fair comparison) with ETHOS, which, despite limitations in temporal modeling, is one of the few approaches capable of generating temporal information. Since ETHOS outputs time range tokens, timestamps were sampled and rounded to the nearest 5 minutes to match the reso-

```
# [ Ours (Value share) ]          # [ ETHOS ]
Birth : 2055. 03. 02              Birth : 2055. 03. 02
Sex: Female                      Sex: Female
Ethnicity: Asian                 Ethnicity: Asian
Age: 81                          Age: 81

2136-10-02 13:25:04              2136-10-02 13:25:04
 - ICU transfer                   - ICU transfer
2136-10-02 13:49:59              2136-10-02 13:49:59
 - RR : 12                        - RR : 12
2136-10-02 14:00:00              2136-10-02 14:00:00
 - RR : 18                        - RR : 18
 - SBP : 118                      - SBP : 118
 - O2 saturation : 99             - O2 saturation : 99
 - MBP : 77                       - MBP : 77
 - DBP : 51                       - DBP : 51
 - GCS-V : 1                      - GCS-V : 1
  - GCS : 3                        - GCS : 3
 - GCS-M : 1                      - GCS-M : 5
 - Temperature : 38.3             - GCS-E : 2
 - GCS-E : 1                      - Temperature : 38.2
2136-10-02 14:04:00               - HR : 74
 - Potassium (ER) : 3.5          2136-10-02 14:50:00
 - PO2 : 492.0                    - RR : 15
 - PEEP : 5.0                     - SBP : 96
 - CO2 : 26.0                     - O2 saturation : 100
 - pH : 7.44                      - MBP : 89
 - Base excess : 1.0              - DBP : 50
 - Hemoglobin (ER) : 10.1         - HR : 78
 - PCO2 : 37.0                   2136-10-02 15:35:00
2136-10-02 14:10:00               - RR : 28
 - SBP : 104                      - O2 saturation : 96
 - RR : 15                       2136-10-02 16:45:00
 - O2 saturation : 99             - GCS : 9
 - MBP : 69                       - GCS-E : 1
 - DBP : 45                       - DBP : 80
2136-10-02 15:00:00               - GCS-M : 3
 - SBP : 125                      - O2 saturation : 96
 - GCS : 3                        - MBP : 86
 - O2 saturation : 100            - HR : 76
 - MBP : 76                       - SBP : 139
 - HR : 80 (truncated rest)       - GCS-V : 1 (truncated rest)
```

Figure 4: Given the initial record (orange), the subsequent medical history is generated (blue). PEEP: Positive end-expiratory pressure, ER: emergency lab.

lution of MIMIC-III (only for ETHOS). For both models, binned measurement values were decoded to actual values by sampling from the empirical distributions of the training data.

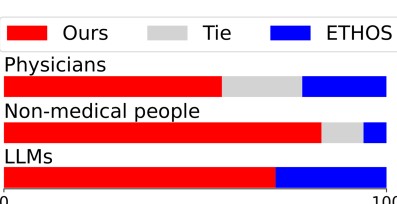

Figure 5: Generated patient EHRs were evaluated by five physicians and five non-medical participants and four LLMs, with 100 comparison responses collected for each category.

Figure 4 presents generated medical history sequences from our model and ETHOS, given the same initial EHR records. From a content perspective, the Glasgow Coma Scale (GCS) should equal the sum of GCS-E/V/M. At 10-02 14:00, our model generated E:1, V:1, M:1; GCS:3, correctly capturing this relationship, whereas ETHOS produced E:2, V:1, M:5; GCS:3, which is inconsistent. Moreover, our model reflected early emergency labs and a variety of tests, followed by routine vital sign checks, while ETHOS generated no labs. From a temporal perspective, our model first performed several tests at short intervals after admission, then naturally returned to an hourly routine. In contrast, ETHOS produced events at irregular intervals and often failed to follow the typical hourly schedule.

We further evaluated 100 generated samples with three evaluator groups: physicians (n=5), non-medical participants (n=5), and commercial LLMs (n=4; ChatGPT (via API, accessed Sep 2025), Gemini 2.5 Flash (via API), 2.5 Pro (via API), Claude 4 Sonnet (via API)). After reviewing up to 10 ground-truth EHR samples, each group assessed subsequent EHR records generated from the same initial records. Figure 5 shows that our model consistently outperformed ETHOS. The LLM input prompts and the generated samples are presented in Appendix F.2.

To verify whether our model effectively integrates calendrical information, we generated vital signs conditioned on time across a 24-hour window (00:00–24:00) based on the same patient history. Figure 6 illustrates that our model generated higher heart rate and temperature values during daytime hours, reflecting realistic circadian variation. In contrast, ETHOS, even for the control variable Height, produced clinically implausible patterns across all cases.

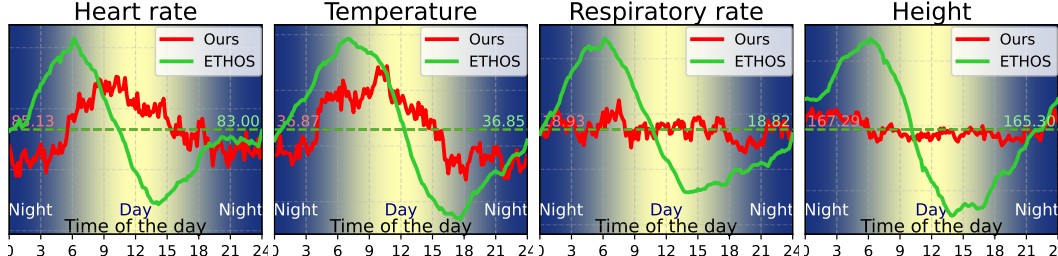

Figure 6: Assessment of the model's ability to capture calendrical temporal patterns. Heart rate, body temperature, and respiratory rate are physiologically higher during the day and lower at night. Using these three features along with height as a control, we let the model sequentially generate predictions across 00:00–24:00, averaged over 1,000 test samples.

## 5 CONCLUSION

We present a novel approach for modeling the unique characteristics of Electronic Health Record (EHR) data, including irregular time intervals and complex numerical values. This work introduces three key contributions: Pathology-Focused Binning to emphasize clinically significant numerical ranges, Dual-Calendar Rotary Position Embedding (RoPE) to encode relative and absolute calendrical time, and a Time-Conditioned Foreseeing (TCF) training objective. TCF enables temporal generative modeling by predicting future timestamps and forecasting events, reflecting clinical planning. Our model outperforms existing foundation models on seven downstream tasks with up to 48% improvement in AUPRC, while generating realistic and temporally consistent EHRs for long-range clinical forecasting. **Limitations:** There is currently no established metric to evaluate the temporal generative performance of EHR models. Assessing the appropriateness of timing is crucial, making conventional methods used for evaluating LLM generation difficult to apply. Developing quantitative evaluation metrics for EHR generation will be important for advancing EHR foundation models.

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

# Appendix

# A    ADDITIONAL RELATED WORKS

The pursuit of powerful and versatile foundation models for Electronic Health Records (EHRs) has led to a rapid evolution of modeling techniques. Current models drew heavily from advancements in Natural Language Processing (NLP), treating EHR data as sequences of discrete events. However, the unique characteristics of EHRs—specifically their sparse, irregularly sampled nature, and the continuous numerical values—have necessitated the development of more specialized architectures and learning objectives.

### EARLY APPROACHES: MODELING CATEGORICAL EVENT SEQUENCES

The initial wave of EHR foundation models adapted the successful Transformer architecture from NLP to the clinical domain. These models primarily focused on learning representations from sequences of medical codes, such as diagnoses, procedures, and medications, while largely omitting numerical and temporal data.

*BEHRT* (Li et al., 2020) (Transformer for Electronic Health Records) introduced the use of the Bidirectional Encoder Representations from Transformers (BERT (Devlin et al., 2019)) architecture for EHR data. It treats a patient's EHR as a sequence of "sentences," where each sentence is a collection of medical codes from a single visit. *BEHRT* is pre-trained on a large dataset of patient records using a Masked Language Model (MLM) objective, where the model learns to predict masked medical codes based on their context. An additional task, Next Visit Prediction (NVP), was also used to predict the codes for a subsequent visit. While effective for tasks like disease prediction, *BEHRT*'s exclusion of numerical values and timestamps limits its clinical utility, as it cannot capture disease severity or the precise timing of events.

*Med-BERT* (Rasmy et al., 2021) followed a similar approach to *BEHRT*, applying the BERT architecture to structured EHR data. It also represents patient histories as sequences of medical codes and uses an MLM pre-training objective to learn contextualized embeddings. *Med-BERT* demonstrated strong performance on various downstream tasks, including disease prediction and patient mortality prediction. However, like *BEHRT*, it does not explicitly model the temporal intervals between visits or the continuous values of lab tests, which are crucial for a comprehensive understanding of a patient's health trajectory.

*EHR-BERT* (Niu et al., 2024) is another BERT-based model that focuses on detecting anomalies in EHR data. It learns the typical patterns of medical events and flags deviations from these patterns as potential anomalies. While its primary application is in data quality and fraud detection, it shares the same fundamental limitations as other early BERT-based models in its handling of EHR data, as it does not incorporate numerical or temporal information into its core architecture.

Further including code-based models: ClinicalMamba (Yang et al., 2024), and Foresight (Kraljevic et al., 2024), these models established the viability of large-scale pre-training for EHR data and demonstrated the power of the Transformer architecture in capturing the complex relationships between medical events. However, their reliance on a purely categorical representation of patient histories highlighted the need for more sophisticated methods that could incorporate the rich numerical and temporal information present in EHRs.

### INCORPORATING NUMERIC VALUES

Recognizing the limitations of purely categorical models, subsequent research focused on integrating continuous numerical values, such as lab results and vital signs, into the modeling process. A common approach has been to discretize these values into a set of predefined bins, allowing them to be treated as discrete tokens within the existing language modeling framework.

*HEART* (Huang et al., 2024) employs this discretization strategy. They convert numeric values into a fixed number of uniform bins (e.g., 10 bins) and create a unique token for each "feature-value" pair. This allows them to capture the magnitude of numerical measurements to some extent. However, this approach has two major drawbacks. First, it leads to a massive increase in the vocabulary size,

as each feature needs its own set of value-specific tokens. This exacerbates data sparsity issues and increases the model's memory footprint. Second, uniform binning can be suboptimal, as it focuses on the non-significant range of clinical values and may not provide sufficient resolution for clinically significant changes.

*FM4EHR* (Burkhart et al., 2025) (Foundation Models for Electronic Health Records) proposed a more efficient method for handling numeric values. Instead of creating unique tokens for each feature-value pair, *FM4EHR* separates the tokenization of features and values. This allows different features to share the same set of value tokens, significantly reducing the vocabulary size and mitigating the data sparsity problem. This "value sharing" approach is a key innovation that allows for more scalable and efficient modeling of numerical data. However, *FM4EHR* still does not explicitly model the temporal aspect of EHR data, relying on the implicit ordering of events in the sequence.

### ADDRESSING TEMPORAL INFORMATION

The timing of medical events is often as important as the events themselves. Another line of research has focused on developing models that can explicitly receive temporal information of EHR data.

*MOTOR* (Steinberg et al., 2024) (A Time-to-Event Foundation Model for Structured Medical Records) is a model specifically designed for survival analysis and time-to-event prediction. It processes sequences of medical codes and learns to predict the time to a future event of interest. (*Note!* it does not take numerical value) *MOTOR* represents time by discretizing the time horizon into a set of predefined intervals and models the hazard function within each interval. This allows it to capture the temporal dependencies between events and make time-aware predictions. However, *MOTOR*'s primary limitation is that it does not incorporate numerical values, which are often strong predictors of patient outcomes. Additionally, its reliance on predefined time intervals and the assumption of a constant hazard function within each interval can limit its temporal precision.

*STraTS* (Tipirneni & Reddy, 2022) (Self-Supervised Transformer for Sparse and Irregularly Sampled Multivariate Clinical Time-Series) takes a different approach to modeling time and values. It tokenizes only the categorical features and embeds the time intervals and numerical values as continuous variables. *STraTS* is trained using a Mean Squared Error (MSE) loss to predict the values of different features at future time points. This allows it to handle irregularly sampled data and make fine-grained predictions. However, by only predicting the next 2-hour value, *STraTS* loses important contextual information and cannot be used for generative modeling of entire patient trajectories.

*TRADE* (Zhu et al., 2024) (Predicting Risk of Alzheimer's Diseases and Related Dementias with AI Foundation Model on Electronic Health Records) also incorporates numerical values, and it uses a non-uniform binning. It discretizes values into nine bins based on their standard deviation from the mean. This approach is more clinically plausible than uniform binning, as it can better capture extreme values that are often indicative of disease. However, *TRADE* does not employ value sharing, which means it still faces the challenge of a large and sparse vocabulary.

*EhrMamba* (Fallahpour et al., 2025) is a recent model that leverages the Mamba architecture, a type of State Space Model (SSM), to efficiently process long EHR sequences. It tokenizes categorical features and uses uniform binning for numerical values. It uses time2vec module Kazemi et al. (2019) to capture temporal dependencies. The use of the Mamba architecture allows *EhrMamba* to scale to much longer patient histories than Transformer-based models, which have a quadratic complexity with respect to sequence length.

*ETHOS* Renc et al. (2024) (Zero-shot health trajectory prediction using transformer) introduces a novel method for explicitly modeling the time intervals between events. It discretizes the time gaps into 13 logarithmic bins, ranging from minutes to months, and inserts a special "time token" between each event token in the input sequence. This allows the model to explicitly reason about the temporal relationships between events. *ETHOS* also incorporates numerical values through binning and value sharing. While this explicit time tokening is a significant step forward, it can increase the sequence length and computational cost. Moreover, the discretization of time still imposes a limit on the model's temporal precision.

# B ADDITIONAL METHOD

## B.1 PATHOLOGY-FOCUSED BINNING

This numerical value tokenization method is designed to create a granular representation for clinically important pathologic values. This non-parametric approach assigns greater resolution to sparse, low density value ranges without making distributional assumptions. The process consists of two main stages: (1) value weight assignment via Kernel Density Estimation, and (2) weighted percentile binning using these assigned weights.

### B.1.1 VALUE WEIGHT ASSIGNMENT VIA KERNEL DENSITY ESTIMATION

The core principle is to assign low weights to values in high-density (physiologic) regions and high weights to values in low-density (pathologic) regions. This is achieved by estimating the data density for each medical feature and assigning a weight inversely proportional to this density.

For a feature with a set of values $V$ and standard deviation $\sigma$, we define a set of discrete representative points $X = \{x_1, x_2, \ldots, x_M\}$. These points span the feature's range $[\min(V), \max(V)]$ and are spaced at uniform intervals of $0.05\sigma$.

At each discrete point $x_k \in X$, we estimate the data density $\rho(x_k)$ by applying a Gaussian convolution kernel. This is a form of Kernel Density Estimation (KDE), where the density at $x_k$ is the sum of influences from all unique data values $v_j \in V$. The density is:

$$\rho(x_k) = \sum_{j=1}^{|V|} K_h(x_k - v_j)$$

Here, $K_h(u)$ is an unnormalized Gaussian kernel defined as:

$$K_h(u) = \exp\left(-\frac{u^2}{2h^2}\right)$$

The bandwidth $h$, which controls the smoothness of the density estimate, is set to $h = 0.1\sigma$ to capture local variations.

From this density, we calculate a raw weight $w_{\text{raw}}(x_k) = 1/(\rho(x_k) + \epsilon)$ for each discrete point. These weights are then normalized and clipped to produce the final weight:

$$w_{\text{final}}(x_k) = \min\left(\frac{w_{\text{raw}}(x_k)}{\min_{k'} w_{\text{raw}}(x_{k'})}, w_{\text{max}}\right)$$

where $w_{\text{max}}$ is a predefined ceiling (e.g., 10). Finally, each original unique value $v_j$ is assigned the weight of its nearest discrete point, $w(v_j) = w_{\text{final}}(x_{k^*})$, where $k^* = \arg\min_k |v_j - x_k|$.

### B.1.2 WEIGHT CALCULATION AND NORMALIZATION

With weights assigned, we proceed to the binning stage. The goal is to partition the feature's values into $B$ bins such that regions with higher weights are given more bins.

We start by calculating a **weighted count** $c'_j$ for each unique value $v_j$:

$$c'_j = c_j \cdot w(v_j)$$

This new count reflects the value's clinical importance as determined by its rarity. Next, we compute the total weighted count for the feature, $C'_{\text{total}} = \sum_{j=1}^{|V|} c'_j$.

The bin thresholds are then determined from the cumulative distribution of these weighted counts. For a set of sorted unique values $v_1 < v_2 < \cdots < v_{|V|}$, the cumulative weighted count up to value $v_k$ is $S_k = \sum_{j=1}^{k} c'_j$. The threshold for the $p$-th bin (where $p \in \{1, 2, \ldots, B-1\}$) is set to the first value $v_k$ whose cumulative weighted share meets or exceeds the $p/B$ percentile:

$$T_p = \min\{v_k \mid \frac{S_k}{C'_{\text{total}}} \geq \frac{p}{B}\}$$

This procedure ensures that value ranges containing high-weight (pathologic) data contribute more significantly to the cumulative sum. As a result, a smaller span of these values is needed to cross a percentile boundary, leading to a denser allocation of bin thresholds in these clinically important regions.

---

**Algorithm 1** Density-Based Value Weight Assignment

---

**Require:**
    `item_counters`: A map from item ID $\rightarrow$ {value: count}.
    $w_{max}$: Maximum weight threshold (e.g., 10).
**Ensure:**
    `value_weights`: A map from item ID $\rightarrow$ {value: weight}.
1: **Initialize** `value_weights` $\leftarrow \emptyset$
2: **for all** `item_id`, `counter` **in** `item_counters` **do**
3:    $V \leftarrow$ sorted unique values from `counter`
4:    $C \leftarrow$ corresponding counts for each value in $V$
5:    $AllValues \leftarrow$ list of all values repeated by their counts
6:    $\sigma \leftarrow$ StandardDeviation($AllValues$)
7:    **if** $\sigma = 0$ **then**
8:        `item_weights` $\leftarrow \{v : 1.0 \text{ for } v \in V\}$
9:        `value_weights[item_id]` $\leftarrow$ `item_weights`
10:       **continue**
11:   $h \leftarrow 0.1 \times \sigma$                             $\triangleright$ Set bandwidth for the Gaussian kernel
12:   $interval \leftarrow 0.05 \times \sigma$
13:   $SplitPoints \leftarrow$ Generate points from $\min(V)$ to $\max(V)$ with $interval$
                                          $\triangleright$ Step 1: Kernel Density Estimation
14:   `densities` $\leftarrow \emptyset$
15:   **for all** split point $x$ **in** $SplitPoints$ **do**
16:      $\rho(x) \leftarrow \frac{1}{|V|} \sum_{j=1}^{|V|} C_j \cdot \exp\left(-\frac{(x-V_j)^2}{2h^2}\right)$
17:      `densities.append(`$\rho(x)$`)`
                            $\triangleright$ Step 2: Calculate, Normalize, and Clip Weights
18:   $w_{raw} \leftarrow 1.0/(\text{`densities`} + 10^{-10})$
19:   $w_{norm} \leftarrow w_{raw}/\min(w_{raw})$
20:   $w_{final} \leftarrow \text{clip}(w_{norm}, 1.0, w_{max})$
                                    $\triangleright$ Step 3: Assign weights to original values
21:   `item_weights` $\leftarrow \emptyset$
22:   **for all** value $v$ **in** $V$ **do**
23:      $closest\_idx \leftarrow \arg\min_k |SplitPoints_k - v|$
24:      `item_weights[`$v$`]` $\leftarrow w_{final}[closest\_idx]$
25:   `value_weights[item_id]` $\leftarrow$ `item_weights`
26: **return** `value_weights`

---

---

**Algorithm 2** Weighted Percentile Binning

---

**Require:**
    `item_counters`: A map from item ID $\rightarrow$ {value: count}.
    `value_weights`: The output from Algorithm 1.
    $B$: The desired number of bins (e.g., 100).
**Ensure:**
    `bin_thresholds`: A map from item ID $\rightarrow$ a list of $B - 1$ thresholds.
1:  **Initialize** `bin_thresholds` $\leftarrow \emptyset$
2:  **for all** `item_id`, `counter` **in** `item_counters` **do**
3:      $V \leftarrow$ sorted unique values from `counter`
4:      $N \leftarrow |V|$
                                                      $\triangleright$ Apply weights to counts for percentile calculation
5:      **if** `value_weights` is provided **then**
6:         $C' \leftarrow [$`counter`$[v_j] \times$ `value_weights`$[$`item_id`$][v_j]$ for $v_j \in V]$
7:      **else**                                                 $\triangleright$ Uniform binning case
8:         $C' \leftarrow [$`counter`$[v_j]$ for $v_j \in V]$
9:      `thresholds` $\leftarrow$ a list of size $B - 1$
10:     **if** $N \geq B$ **then**
11:        $C'_{total} \leftarrow \sum C'$
12:        $C'_{cumulative} \leftarrow$ CumulativeSum$(C')$
13:        **for** $p = 1$ to $B - 1$ **do**
14:           $target\_count \leftarrow C'_{total} \times p/B$
15:           $idx \leftarrow$ FindFirstIndexWhere$(C'_{cumulative} > target\_count)$
16:           `thresholds`$[p - 1] \leftarrow V[idx + 1]$            $\triangleright$ Handle edge cases
17:      **else**                $\triangleright$ Apply specific assignment for sparse values (centering or striding)
18:        `thresholds` $\leftarrow$ Generate thresholds based on sparse assignment logic
19:      `bin_thresholds`$[$`item_id`$] \leftarrow$ `thresholds`
20: **return** `bin_thresholds`

---

### B.2   DUAL-CALENDAR ROTARY POSITION EMBEDDING

To address the unique temporal characteristics of Electronic Health Record (EHR) data—namely, the highly irregular event intervals and the clinical significance of calendrical time—we propose **Dual-Calendar Rotary Position Embedding (RoPE)**. This method extends the conventional RoPE by partitioning the embedding dimension within each attention head to jointly encode both the **relative sequence order** of tokens and their **absolute calendrical time**.

For a given query or key vector $x \in \mathbb{R}^{d_k}$ in an attention head, we partition it into two subspaces: a positional component $x_{pos} \in \mathbb{R}^{d_{pos}}$ and a temporal component $x_{time} \in \mathbb{R}^{d_{time}}$, where $d_k = d_{pos} + d_{time}$.

$$x = [x_{pos} \parallel x_{time}]$$

Each component is then rotated using a specialized RoPE variant before being concatenated back together.

### B.2.1   POSITIONAL DIMENSION ENCODING

The $x_{pos}$ component, corresponding to the first $d_{pos}$ dimensions, employs the standard RoPE formulation. With a reduced dimensionality of $d_{pos}$, this component does not rescale its rotational frequencies to cover a wide positional range. Instead, it effectively **truncates the frequency spectrum**, retaining the high-frequency rotations corresponding to the initial dimensions of a standard RoPE. This strategic choice is predicated on the observation that its primary role is now to disambiguate the order of co-occurring events that share an identical timestamp. For a token at position $m$, the rotation angle $\theta_{p,i}^{(pos)}$ is defined as:

$$\theta_{p,i}^{(pos)} = \frac{p}{\text{base}^{2i/d}} \ , \ \ i \in \{1, 2, ..., d_{pos}/2\}$$

The task of modeling long-range temporal dependencies is thus naturally offloaded to the Calendar-Time dimension, which is explicitly designed for this purpose.

### B.2.2 CALENDAR-TIME DIMENSION ENCODING

The core novelty of our approach lies in the encoding applied to the $x_{time}$ component. This component is designed to encode an event's absolute timestamp, $t$, by capturing its periodicity across multiple, clinically relevant time scales. This is achieved using a predefined set of **semantically meaningful calendrical periods**, $S = \{s_1, s_2, \ldots, s_{d_{time}/2}\}$, as detailed in Table 5. These periods capture periodicities ranging from short-term diurnal patterns to long-term annual and multi-year trends.

For each period $s_j$ from this set, we calculate a unique rotation angle $\theta_{t,j}^{(time)}$ that represents the phase of the event within that specific period. The formula for the rotation angle is:

$$\theta_{t,j}^{(time)} = \left( \frac{t \pmod{s_j}}{s_j} \right) \cdot 2\pi$$

This mechanism produces a **multi-scale temporal representation**. Events occurring at the same time of day but on different dates will share the exact same rotation for the 'day' period, allowing the model to easily learn periodical patterns.

### B.2.3 INTEGRATION AND APPLICATION

Finally, the two rotated components are concatenated to form the final query and key vectors. The full transformation for a query vector $q = [q_{pos} \| q_{time}]$ to its rotated form $q'$ is:

$$q' = [\text{RoPE}(q_{pos}, \theta^{(pos)}) \| \text{RoPE}(q_{time}, \theta^{(time)})]$$

An identical transformation is applied to the key vector $k$. By equipping the self-attention mechanism with this dual-encoding strategy, our model can simultaneously reason about the sequential flow of information and the absolute, cyclical context of clinical events.

Table 5: Predefined Calendrical Periods for Temporal Encoding

| Category | Period Name | Duration (seconds) |
|---|---|---|
| *Short-term* | 5 minutes | 300 |
| | 10 minutes | 600 |
| | 30 minutes | 1,800 |
| | 1 hour | 3,600 |
| | 3 hours | 10,800 |
| | 12 hours | 43,200 |
| *Mid-term* | 1 day | 86,400 |
| | 2 days | 172,800 |
| | 1 week | 604,800 |
| | 2 weeks | 1,209,600 |
| | 1 month | 2,629,746 |
| | 1 season (3 months) | 7,889,238 |
| | 6 months | 15,778,476 |
| *Long-term* | 1 year | 31,556,952 |
| | 2 years | 63,113,904 |
| | 4 years | 126,227,808 |
| | 10 years | 315,569,520 |
| | 30 years | 946,708,560 |
| | 100 years | 3,155,695,200 |
| | 300 years | 9,467,085,600 |

### B.3 TIME CONDITIONED FORESEE OBJECTIVE

To elucidate the mechanics of our proposed Time-Conditioned Foresee (TCF) module, we provide a step-by-step explanation. This description follows the flow of information from the initial input—the

final hidden state of a backbone model—to the module's dual outputs. The detailed computational flow is presented in Algorithm 3.

We define the primary dimensions:

- $B$ denotes the batch size.
- $L$ denotes the sequence length.
- $d_{\text{model}}$ represents the hidden dimension of the backbone model's output.
- $d_{\text{embed}}$ is the dimensionality of our internal time embeddings, set to 32.
- $N_{\text{scales}}$ is the number of time scales used for decomposition, 11 in our implementation.
- $N_{\text{foresee}}$ is the number of future timestamps provided for the foresee objective, set to 10.
- $C_i$ is the number of discrete categories for the $i$-th time scale.

The process begins with the final hidden state from the backbone decoder for each token in the sequence.

**Input:** The last hidden state tensor, $H_{\text{last}}$.

- $H_{\text{last}} \in \mathbb{R}^{B \times L \times d_{\text{model}}}$

### B.3.1  A. NEXT EVENT TIME PREDICTION PATH ($\mathcal{L}_{\text{NEXT\_TIME}}$)

This pathway is responsible for predicting the time until the next event.

**1. Initial Projection**  The input hidden state $H_{\text{last}}$ is passed through a two-layer Feed-Forward Network (FFN), denoted as $\text{FFN}_{\text{enc}}$, to create a representation for time prediction.

- **Input:** $H_{\text{last}} \in \mathbb{R}^{B \times L \times d_{\text{model}}}$
- **Output:** An intermediate time-focused tensor, $H_{\text{time}}$.
    - $H_{\text{time}} = \text{FFN}_{\text{enc}}(H_{\text{last}}) \in \mathbb{R}^{B \times L \times (N_{\text{scales}} \cdot d_{\text{embed}})}$

**2. Logit Generation**  The tensor $H_{\text{time}}$ is conceptually partitioned into $N_{\text{scales}}$ segments. Each segment is used to compute the logits for its corresponding time scale by multiplying it with the respective time embedding weight matrix.

- **Input:** $H_{\text{time}}$, treated as $N_{\text{scales}}$ tensors $\{H_{\text{time}}^{(i)}\}_{i=1}^{N_{\text{scales}}}$, where each $H_{\text{time}}^{(i)} \in \mathbb{R}^{B \times L \times d_{\text{embed}}}$.
- **Operation:** For each scale $i$, we compute logits: $\text{Logits}^{(i)} = H_{\text{time}}^{(i)} \cdot (W_{\text{embed}}^{(i)})^T$, where $W_{\text{embed}}^{(i)} \in \mathbb{R}^{C_i \times d_{\text{embed}}}$.
- **Output:** A set of $N_{\text{scales}}$ logit tensors.
    - $\text{Logits}^{(i)} \in \mathbb{R}^{B \times L \times C_i}$

**3. Ground-Truth Label Decomposition**  To compute the loss, these logits are compared against ground-truth labels. Instead of regressing a continuous time value, we transform the ground-truth time delta (in seconds) into a set of categorical integer labels. This is achieved through a deterministic process analogous to a mixed-radix conversion, using the time scales defined in Table 6.

For instance, assume a ground-truth time delta $\Delta T_{\text{next}}$ of **34,586,130 seconds**. The conversion to a vector of $N_{\text{scales}}$ integer labels proceeds sequentially from the largest time scale to the smallest, using integer division to find the label and the modulo operator to find the remainder for the next step.

1. **`year10`:** $34,586,130 \,//\, 315,360,000 = 0$. Remainder: $34,586,130$. $\rightarrow$ **Label: 0**
2. **`year1`:** $34,586,130 \,//\, 31,536,000 = 1$. Remainder: $3,050,130$. $\rightarrow$ **Label: 1**
3. **`month3`:** $3,050,130 \,//\, 7,948,800 = 0$. Remainder: $3,050,130$. $\rightarrow$ **Label: 0**
4. **`month1`:** $3,050,130 \,//\, 2,678,400 = 1$. Remainder: $371,730$. $\rightarrow$ **Label: 1**

Table 6: Time scales for multi-scale decomposition, along with their duration in seconds and the number of categories for classification.

| Time Scale | Duration in Seconds | Num. of Categories |
|---|---|---|
| year10 | 315,360,000 | 10 |
| year1 | 31,536,000 | 10 |
| month3 | 7,948,800 | 4 |
| month1 | 2,678,400 | 3 |
| week1 | 604,800 | 5 |
| day1 | 86,400 | 7 |
| hour6 | 21,600 | 4 |
| hour1 | 3,600 | 6 |
| minute10 | 600 | 6 |
| minute1 | 60 | 10 |

5. **week1:** $371,730 // 604,800 = 0$. Remainder: $371,730$. $\rightarrow$ **Label: 0**

6. **day1:** $371,730 // 86,400 = 4$. Remainder: $27,330$. $\rightarrow$ **Label: 4**

7. **hour6:** $27,330 // 21,600 = 1$. Remainder: $5,730$. $\rightarrow$ **Label: 1**

8. **hour1:** $5,730 // 3,600 = 1$. Remainder: $2,130$. $\rightarrow$ **Label: 1**

9. **minute10:** $2,130 // 600 = 3$. Remainder: $330$. $\rightarrow$ **Label: 3**

10. **minute1:** $330 // 60 = 5$. Remainder: $30$. $\rightarrow$ **Label: 5**

11. **position:** A mechanism to account for events occurring simultaneously at the same time. For the next event label, it is set to 0; for subsequent foresee labels, it is set to +1 if the time is the same as the previous one, and 0 otherwise. $\rightarrow$ **Label: 0**

Ultimately, the continuous value of 34,586,130 seconds is converted into the following vector of ten integer labels, which constitutes the ground-truth $Y_{\text{next\_time}}^{(i)}$ for this example:

$$[0, 1, 0, 1, 0, 4, 1, 1, 3, 5, 0]$$

By training the model to predict these categorical labels for each time scale, we transform a difficult regression task into a series of more stable and effective classification tasks. The final scalar loss, $\mathcal{L}_{\text{next\_time}}$, is the average Cross-Entropy loss calculated between the generated logits and these decomposed ground-truth labels.

### B.3.2   B. TIME-CONDITIONING PATH FOR FORESEE OBJECTIVE

This pathway conditions the hidden state on a set of specified future timestamps to predict upcoming events.

**1. Input Foresee Timestamps**   The module receives future time deltas from the $N_{\text{foresee}}$ future events relative to the current timestamp at each position.

- **Input:** A tensor of future time deltas, $\Delta T_{\text{foresee}} \in \mathbb{Z}^{B \times L \times N_{\text{foresee}}}$.

**2. Time Embedding**   Each time delta in $\Delta T_{\text{foresee}}$ is decomposed into $N_{\text{scales}}$ integer labels (as demonstrated above). These labels are used to look up corresponding vectors from the embedding tables, $\{W_{\text{embed}}^{(i)}\}_{i=1}^{N_{\text{scales}}}$, which are then concatenated. *Note,* we share the weights for embedding and unembedding timestamps.

- **Input:** Decomposed time labels, $C_{\text{foresee}} \in \mathbb{Z}^{B \times L \times N_{\text{foresee}} \times N_{\text{scales}}}$.

- **Output:** A dense time embedding tensor, $E_{\text{time}}$.
    - $E_{\text{time}} \in \mathbb{R}^{B \times L \times N_{\text{foresee}} \times (N_{\text{scales}} \cdot d_{\text{embed}})}$

**3. Projection of Time Embedding**    The concatenated embedding $E_{\text{time}}$ is projected to the model's hidden dimension via $\text{FFN}_{\text{dec}}$.

- **Input:** $E_{\text{time}} \in \mathbb{R}^{B \times L \times N_{\text{foresee}} \times (N_{\text{scales}} \cdot d_{\text{embed}})}$

- **Output:** A processed time conditioning tensor, $H_{\text{time\_cond}}$.

    - $H_{\text{time\_cond}} = \text{FFN}_{\text{dec}}(E_{\text{time}}) \in \mathbb{R}^{B \times L \times N_{\text{foresee}} \times d_{\text{model}}}$

**4. Time-Conditioning via Fusion**    The final step fuses the original hidden state $H_{\text{last}}$ with the processed time conditioning tensor $H_{\text{time\_cond}}$. To align their dimensions for the element-wise addition, $H_{\text{last}}$ is first expanded by inserting a new dimension. This prepares it for broadcasting across the $N_{\text{foresee}}$ dimension, allowing each of the $N_{\text{foresee}}$ time embeddings to condition the single original hidden state.

- **Inputs:**

    - Original hidden state: $H_{\text{last}} \in \mathbb{R}^{B \times L \times d_{\text{model}}}$

    - Time conditioning tensor: $H_{\text{time\_cond}} \in \mathbb{R}^{B \times L \times N_{\text{foresee}} \times d_{\text{model}}}$

- **Operation:** The fusion is performed via a residual connection. First, $H_{\text{last}}$ is unsqueezed, and then added to $H_{\text{time\_cond}}$.

    - $H'_{\text{last}} = \text{Unsqueeze}(H_{\text{last}}, \dim = 2) \in \mathbb{R}^{B \times L \times 1 \times d_{\text{model}}}$

    - $H_{\text{fused}} = H'_{\text{last}} + H_{\text{time\_cond}}$    *// Broadcasting occurs along the $N_{foresee}$ dimension.*

- **Output:** The final time-conditioned hidden state, $H_{\text{conditioned}}$, after Layer Normalization and a final FFN block.

    - $H_{\text{conditioned}} \in \mathbb{R}^{B \times L \times N_{\text{foresee}} \times d_{\text{model}}}$

SUMMARY OF MODULE OUTPUTS

The TCF module produces two primary outputs:

1. **Next Time Loss** ($\mathcal{L}_{\textbf{next\_time}}$): A scalar value for backpropagation.

2. **Conditioned Hidden State** ($H_{\textbf{conditioned}}$): A tensor of shape $\mathbb{R}^{B \times L \times N_{\text{foresee}} \times d_{\text{model}}}$, which serves as the input to the final prediction head for calculating the foresee loss, $\mathcal{L}_{\text{foresee}}$. The ground-truth label corresponding to each conditioned hidden vector is the `Feature` token of the actual clinical event that occurred at the given foresee timestamp.

---

**Algorithm 3** Time-Conditioned Foresee (TCF) Module

---

**Require:**
1: $H_{\text{last}} \in \mathbb{R}^{B \times L \times d_{\text{model}}}$: Last hidden states from the backbone model.
2: $T_{\text{current}} \in \mathbb{R}^{B \times L}$: Absolute timestamps for each hidden state in $H_{\text{last}}$.
3: $T_{\text{next}} \in \mathbb{R}^{B \times L}$: Absolute timestamps of the next event for each position.
4: $T_{\text{foresee}} \in \mathbb{R}^{B \times L \times N_{\text{foresee}}}$: A set of absolute future timestamps for conditioning.
5: $\mathcal{W}$: All trainable weights, including FFNs and embedding tables $\{W_{\text{embed}}^{(i)}\}_{i=1}^{N_{\text{scales}}}$.
6: $Periods$: A dictionary mapping each time scale to its duration in seconds.

**Ensure:**
7: $\mathcal{L}_{\text{next\_time}} \in \mathbb{R}$: The loss for next event time prediction.
8: $H_{\text{conditioned}} \in \mathbb{R}^{B \times L \times N_{\text{foresee}} \times d_{\text{model}}}$: Hidden states conditioned on $T_{\text{foresee}}$.

---

9: **function** TCF_MODULE($H_{\text{last}}, T_{\text{current}}, T_{\text{next}}, T_{\text{foresee}}, \mathcal{W}, Periods$)
            ▷ **Part A: Next Event Time Prediction**
10:      $\Delta T_{\text{next}} \leftarrow T_{\text{next}} - T_{\text{current}}$
11:      $Y_{\text{next}} \leftarrow$ DECOMPOSETIME($\Delta T_{\text{next}}, Periods$)
12:      $H_{\text{time}} \leftarrow$ FFN$_{\text{enc}}(H_{\text{last}})$                       ▷ Shape: $(B, L, N_{\text{scales}} \cdot d_{\text{embed}})$
13:      $H_{\text{time}} \leftarrow$ RESHAPE($H_{\text{time}}, (B, L, N_{\text{scales}}, d_{\text{embed}})$)
14:      $\mathcal{L}_{\text{total}} \leftarrow 0$
15:      **for** $i = 1 \rightarrow N_{\text{scales}}$ **do**
16:          $H_{\text{time}}^{(i)} \leftarrow H_{\text{time}}[:, :, i, :]$                            ▷ Shape: $(B, L, d_{\text{embed}})$
17:          Logits$^{(i)} \leftarrow H_{\text{time}}^{(i)} \cdot (W_{\text{embed}}^{(i)})^T$
18:          $Y_{\text{next}}^{(i)} \leftarrow Y_{\text{next}}[:, :, i]$
19:          $\mathcal{L}_{\text{total}} \leftarrow \mathcal{L}_{\text{total}} +$ CROSSENTROPYLOSS(Logits$^{(i)}, Y_{\text{next}}^{(i)}$)
20:      $\mathcal{L}_{\text{next\_time}} \leftarrow \mathcal{L}_{\text{total}}/N_{\text{scales}}$
            ▷ **Part B: Time-Conditioning for Foresee Objective**
21:      $T_{\text{current}}' \leftarrow$ UNSQUEEZE($T_{\text{current}}, \dim = 2$)              ▷ Shape: $(B, L, 1)$
22:      $\Delta T_{\text{foresee}} \leftarrow T_{\text{foresee}} - T_{\text{current}}'$
23:      $C_{\text{foresee}} \leftarrow$ DECOMPOSETIME($\Delta T_{\text{foresee}}, Periods$)     ▷ Shape: $(B, L, N_{\text{foresee}}, N_{\text{scales}})$
24:      $E_{\text{time\_list}} \leftarrow []$
25:      **for** $i = 1 \rightarrow N_{\text{scales}}$ **do**
26:          $C_{\text{foresee}}^{(i)} \leftarrow C_{\text{foresee}}[:, :, :, i]$
27:          $E_{\text{time}}^{(i)} \leftarrow$ LOOKUP($W_{\text{embed}}^{(i)}, C_{\text{foresee}}^{(i)}$)            ▷ Shape: $(B, L, N_{\text{foresee}}, d_{\text{embed}})$
28:          Append $E_{\text{time}}^{(i)}$ to $E_{\text{time\_list}}$
29:      $E_{\text{time}} \leftarrow$ CONCATENATE($E_{\text{time\_list}}, \dim = -1$)     ▷ Shape: $(B, L, N_{\text{foresee}}, N_{\text{scales}} \cdot d_{\text{embed}})$
30:      $H_{\text{time\_cond}} \leftarrow$ FFN$_{\text{dec}}(E_{\text{time}})$           ▷ Shape: $(B, L, N_{\text{foresee}}, d_{\text{model}})$
31:      $H_{\text{last}}' \leftarrow$ UNSQUEEZE($H_{\text{last}}, \dim = 2$)            ▷ Shape: $(B, L, 1, d_{\text{model}})$
32:      $H_{\text{fused}} \leftarrow$ LAYERNORM($H_{\text{last}}' + H_{\text{time\_cond}}$)
33:      $H_{\text{conditioned}} \leftarrow$ LAYERNORM($H_{\text{fused}} +$ FFN$_{\text{final}}(H_{\text{fused}})$)
34:      **return** $\mathcal{L}_{\text{next\_time}}, H_{\text{conditioned}}$

35: **function** DECOMPOSETIME($\Delta T, Periods$)        ▷ Helper function for time decomposition
36:      $R \leftarrow \Delta T$
37:      $Labels \leftarrow []$
38:      **for** scale in REVERSED($Periods$.keys()) **do**
39:          $L_{\text{scale}} \leftarrow R \, // \, Periods[\text{scale}]$                        ▷ Integer division
40:          $R \leftarrow R \, \% \, Periods[\text{scale}]$                         ▷ Modulo operation
41:          $L_{\text{scale}} \leftarrow$ CLAMP($L_{\text{scale}}, \min = 0, \max = C_{\text{scale}} - 1$)
42:          Prepend $L_{\text{scale}}$ to $Labels$
43:      **return** STACK($Labels$)

---

## C  DATA AND PREPROCESSING

### C.1  DATASET: MIMIC-III

MIMIC-III (Johnson et al., 2016a;b) is an openly accessible resource that contains de-identified clinical data from more than 40,000 individuals admitted to the intensive care units of Beth Israel Deaconess Medical Center between 2001 and 2012. The dataset encompasses a wide range of information, including patient demographics, hourly vital sign recordings, laboratory measurements, administered treatments and procedures, prescribed medications, clinical notes, radiology reports, and outcomes such as in-hospital and post-discharge mortality.

The MIMIC-III database was de-identified in compliance with Health Insurance Portability and Accountability Act (HIPAA) standards through data cleansing and systematic date shifting. To preserve clinical intervals, patient-specific dates were consistently shifted into the future by a random offset, placing admissions within the years 2100–2200 while retaining the original time of day, weekday, and approximate seasonality. For patients older than 89, dates of birth were modified such that their recorded ages exceed 300 years, thereby masking their true age in accordance with HIPAA requirements. This modification provides a suitable framework for our Dual-Calendar RoPE, which is designed to address calendrical time, to operate effectively.

### C.2  FEATURE SELECTION

Electronic Health Records (EHRs) are rich with events that have a numerical `Value`, a characteristic that distinguishes them from natural language. Consequently, our experiments focused on events that possess a numerical `Value`. We utilized 17 vital sign features from the `CHARTEVENTS.csv` file, following the selection in Harutyunyan et al. (2019), and the top 100 most frequently measured laboratory tests from `LABEVENTS.csv` as the events for our study. These features are detailed in Table 7. In addition, patient events such as "hospital admission", "ICU transfer (ICU in)", "ICU discharge (ICU out)", and "hospital discharge" were utilized.

### C.3  FURTHER PREPROCESSING

In addition to the preprocessing of Harutyunyan et al. (2019), we made the following modifications: (1) removed outlier values in the laboratory data based on independent evaluations by three physicians and standardized the measurement units; (2) excluded hospitalization episodes with fewer than 10 events; and (3) added an anchor token with a timestamp of January 1st, 00:00 of the same year before each admission token to serve as a calendrical time reference. As a result, the lengths of patients' medical histories follow the distribution shown in Figure 7.

## D  BACKBONE, BASELINES, PRE-TRAINING DETAIL

### D.1  BACKBONE ARCHITECTURE

The backbone model used in this study follows a standard **decoder-only transformer** architecture. To minimize performance variations caused by differences in backbone models and to quantitatively assess the effectiveness of our proposed training methodology, we used the same backbone across all experiments. However, for models trained with Transformer encoders using the masked language modeling approach (HEART, TRADE), we removed the causal mask during pre-training so that they could be used as encoder models. The backbone details are as follows:

- **Vocabulary Size**: 166 (1219 if not share bin)
- **Embedding and Hidden Dimension** ($d_{\text{model}}$): 512
- **Number of Decoder Layers** ($N$): 6
- **Number of Attention Heads**: 8
- **Dimension per Head**: 64
- **Dimension of K, Q**: 64 (Ours: first 24: positional RoPE / last 40: calendrical time RoPE)

Table 7: Selected features from MIMIC-III used in this study.

| Category | Features |
| --- | --- |
| **Ethnicity** - # 10 (from `ADMISSIONS.csv`) | White, White - Russian, White - other European, Asian, Asian - Chinese, Hispanic or Latino, Hispanic/Latino - Dominican, Black/Cape Verdean, Black/African American, Others or Unknown |
| **Vital Signs** - # 17 (from `CHARTEVENTS.csv`) | Capillary refill rate (CRR), Systolic blood pressure (SBP), Mean blood pressure (MBP), Diastolic blood pressure (DBP), Fraction of inspired oxygen (FiO2), Heart rate (HR), Respiratory rate (RR), Glasgow coma scale eye response (GCS-E), Glasgow coma scale motor response (GCS-M), Glasgow coma scale verbal response (GCS-V), Glasgow coma scale (GCS), Serum glucose, O2 saturation, Blood pH, Body temperature, Height, Weight. |
| **Laboratory Tests** - # 100 (from `LABEVENTS.csv`) | Hematocrit, Potassium, Sodium, Creatinine, Chloride, Blood urea nitrogen, Bicarbonate, Platelets, Anion gap, White blood cell count, Hemoglobin chemistry, Mean corpuscular hemoglobin concentration, Red blood cell count, Mean corpuscular hemoglobin, Mean corpuscular volume, Red Cell Distribution Width, Magnesium, Calcium Total, Phosphate, Base excess, CO2 (ETCO2, PCO2, etc.), Partial pressure of oxygen, Partial pressure of carbon dioxide, Partial thromboplastin time, Prothrombin time INR, Prothrombin time, Calcium Free, Bilirubin Total, Alanine aminotransferase, Asparate aminotransferase, Alkaline phosphate, Potassium blood gas, Lactate, Lymphocytes, Neutrophils, Monocytes, Eosinophils, Basophils, Albumin, Creatine Kinase, Oxygen blood gas, Urine Specific Gravity, Creatine Kinase-MB, Lactate dehydrogenase, Urine Protein, Urine Urobilinogen, Urine Ketone, Urine Color, Urine Appearance, Urine Blood, Urine Bilirubin, Urine Nitrite, Urine Leukocyte, Hematocrit blood gas, Hemoglobin blood gas, Troponin-T, Positive end-expiratory pressure, Urine Yeast, Urine White blood cell count, Urine Red blood cell count, Urine Epithelial cells, Band Neutrophils, Urine Bacteria, Sodium blood gas, Lipase, Amylase, Estimated GFR, Hypochromia, Anisocytosis, Macrocytosis, Lymphocytes Atypical, Metamyelocytes, Myelocytes, Microcytes, Poikilocytosis, Vancomycin (blood), Chloride blood gas, Polychromasia, Functional Fibrinogen, Bilirubin Direct, Bilirubin Indirect, Platelet Smear, Urine Creatinine, Thyroid-stimulating hormone, Urine Sodium, Triglycerides, Granulocyte count, CK-MB Index, Phenytoin (blood), Alveolar-arterial gradient, Cholesterol Total, Urine osmolality, Osmolality, Uric acid, Cholesterol HDL, Iron, Cholesterol ratio Total/HDL, Ferritin, Transferrin, Iron binding capacity, HbA1C, Nucleated red cells, Cholesterol, Ovalocyte, Urine Hyaline casts, Urine mucous, Cortisol, Urine urea nitrogen, Haptoglobin, Protein (Total), Vitamin B12, Benzodiazepine Screen, Barbiturate Screen, Tricyclic Antidepressant Screen, Troponin-I, Urine potassium, Tacrolimus level, Schistocytes, Reticulocyte count, Ethanol, Urine Chloride, Acetaminophen, Urine Cocaine, Urine Benzodiazepine screen, Urine Amphetamine screen, Urine Opiate screen, Urine Barbiturate screen, Urine Methadone, Bicarbonate blood gas, Salicylate, Urine Total protein, Teardrop cells, Cyclosporin, Folate, Burr cells, Sedimentation rate, Digoxin, Thyroxine, Globulin, Urine protein/creatine ratio, NT-proBNP, Urine Amorphous cristal, C-reactive protein, Large platelets, Urine Granular casts, Gentamicin, Target cell, Transitional epithelial cells, Fibrin degradation, CSF Lymphs. |

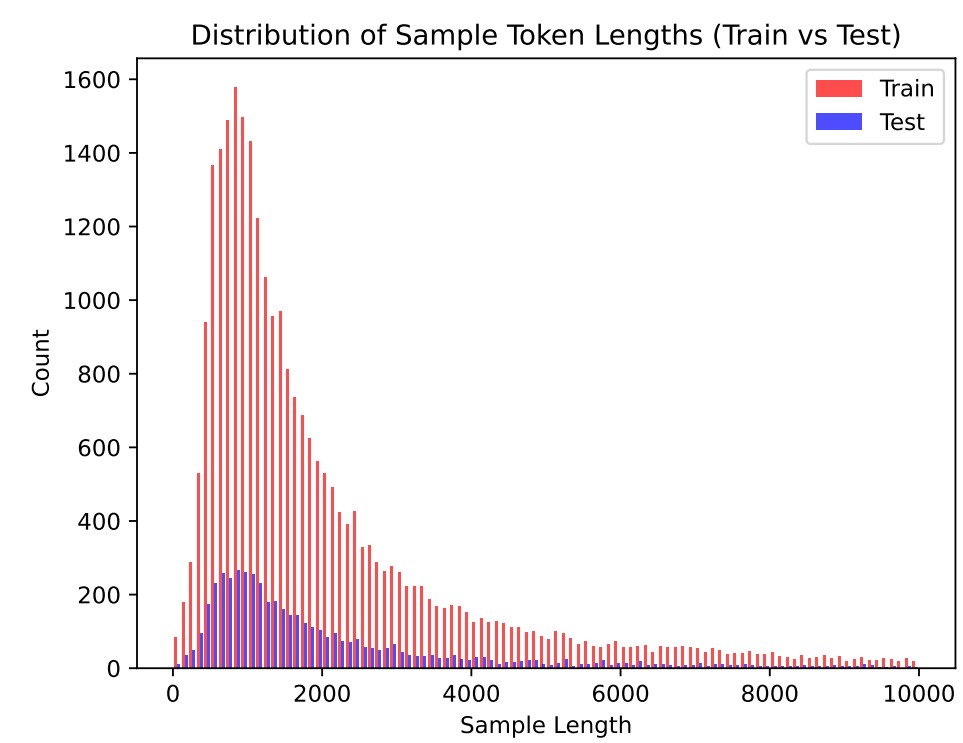

Figure 7: Length distribution of tokenized histories per patient. Most are within 2048 tokens. Lengths exceeding 10k tokens are not shown.

- **Feed-Forward Network (FFN) Inner Dimension ($d_{\mathbf{ff}}$)**: 2048
- **Total Parameters (Backbone)**: 19,001,344 (19,060,736 if not share bin)
- **Activation**: ReLU (Agarap, 2018)

Each decoder layer is composed of two main sub-layers: a **multi-head self-attention** block and a **feed-forward network**. For training stability, the model adopts a **Pre-Layer Normalization (Pre-LN)** structure, where Layer Normalization is applied to the input of each sub-layer. A **residual connection** is then employed around each of the two sub-layers.

### D.2    BASELINE MODELS

We compared our model with baselines that utilized either Time or Value in their training: HEART, FM4EHR, MOTOR, STraTS, EHRSHOT, TRADE, EHRMamba, and ETHOS. The backbone for each model was standardized as described above, while other methodologies (binning, tokenization, positional embedding, learning objective, etc.) followed their original papers. The reproduction details are as follows.

**HEART**    (Total Parameters: 20,204,180)

- **Backbone**: Uses a Transformer encoder due to its MLM-based loss.
- **Bin Sharing**: No, uses Value-Feature paired tokenization.
- **Binning**: 10-uniform binning.
- **Positional Embedding**: Absolute positional embedding—a learned positional embedding of the visit index (0 for the patient's first visit, 1 for the second, and so on).

- **Learning Objective**: For each visit, event tokens (Value-Feature paired) are masked with a probability of $p_{\mathrm{mask}} = 0.15$. The model is trained via a multi-class classification loss to predict the masked token. This is trained separately for different event types (V/S, Lab). Additionally, for unmasked events, values are altered with a probability of $p_{\mathrm{anomaly}} = 0.05$, and the model is trained with a binary classification loss to identify whether the value at each position was altered.

**FM4EHR**  (Total Parameters: 19,001,536)

- **Backbone**: Transformer decoder.
- **Bin Sharing**: Yes, Feature and Value are tokenized separately. Numeric values share bin tokens across different features.
- **Binning**: 10-uniform binning.
- **Positional Embedding**: Uses rotary positional embedding based only on position, without explicit time information.
- **Learning Objective**: Uses NTP loss.

**MOTOR**  (Total Parameters: 20,341,952)

- **Backbone**: Transformer decoder.
- **Methodology Adaptation**: This model was originally designed for feature-only events (e.g., diagnosis codes). To adapt it for continuous values, we set the "occurrence of an abnormal measurement result" as the endpoint for its time-to-event loss. An abnormal value was defined by either being outside the medical normal range (per Harrison's Principles of Internal Medicine (Kasper et al., 2015), evaluated by three clinicians) or being a statistical outlier (e.g., top/bottom 5%).
- **Bin Sharing**: No, uses Value-Feature paired tokenization.
- **Binning**: 10-uniform binning was applied to the Value.
- **Positional Embedding**: The model converts each event's timestamp into 'days since birth' and applies this value in its rotary positional embedding.
- **Learning Objective**: Uses the Time-to-event loss from the original paper.

**STraTS**  (Total Parameters: 19,374,801)

- **Architecture**: As this is an older paper, its structure is not suitable for parallel training. We therefore modified the architecture while retaining the core ideas. We used the last hidden state of the decoder at each time step as the event embedding, instead of the original Fusion Self-Attention mechanism.
- **Value and Time Embedding**: This model does not use binning. Instead, it embeds Features via a look-up table and continuously embeds Value and Time (in hours) via a 2-layer fully connected layer. The resulting embeddings are summed to form the final event embedding.
- **Positional Embedding**: No additional positional embedding is used beyond the Time information included in the event embedding.
- **Demographics**: Unlike typical models, STraTS encodes demographic information (gender, race, age) with a separate MLP and concatenates it to the last hidden state.
- **Learning Objective**: The final embedding is used to predict the value of events occurring within two hours of each event, trained with a mean squared error loss.

**EHRSHOT**  (Total Parameters: 20,278,892)

- **Backbone**: Transformer decoder.
- **Bin Sharing**: No, uses Value-Feature paired tokenization.
- **Binning**: 10-uniform binning.

- **Positional Embedding**: Following CLMBR-T Steinberg et al. (2021), it uses Rotary positional embedding based on position order. Time information is not used beyond ordering for this component.

- **Time Information**: A 5-dimensional time vector is concatenated to the token embedding vector. This vector consists of the z-normalized values of $[\text{age}, \log(\text{age}), \text{days since admission}, \log(\text{days since admission}), \text{first admission indicator}]$. The final concatenated vector length is 512.

- **Learning Objective**: Next token prediction modeling.

**TRADE**   (Total Parameters: 21,617,664)

- **Backbone**: Uses a Transformer encoder due to its MLM-based loss.

- **Bin Sharing**: No, uses Value-Feature paired tokenization.

- **Binning**: 9-standard deviation-based binning. For each feature's value distribution, thresholds are set by adding $\{-10, -3, -1, -0.5, 0.5, 1, 3, 10\}$ standard deviations to the mean, creating 9 bins. This method can be sensitive to outliers, so additional clipping was performed on 17 vital sign data points based on physician guidelines. The clipping ranges are as follows: CRR:[0,1], SBP:[0,400], MBP:[0,300], DBP:[0,300], FiO2:[0,1], HR:[0,200], RR:[0,100], GCS-E:[1,4], GCS-M:[1,6], GCS-V:[1,5], Glucose:[0,1200], O2 saturation:[0,100], Body temperature:[20, 45], Height:[0,1000], Weight:[0,1000].

- **Positional Embedding**: Absolute positional embedding—performs learned positional embedding using three types of sequential and temporal information: 1) The index of the current hospital admission, 2) The number of days passed since admission, and 3) The current age. These are all integers, passed through an embedding layer, and then summed.

- **Learning Objective**: Uses a standard MLM methodology, masking each token with $p = 0.2$ and using the last hidden state of the masked position to predict the pre-mask label via a classification loss.

**EHRmamba**   (Total Parameters: 21,770,752)

- **Backbone**: Although the original paper uses Mamba, we applied the same Transformer decoder backbone for a fair comparison.

- **Bin Sharing**: No, uses Value-Feature paired tokenization.

- **Binning**: 10-uniform binning (the original paper used 5-uniform binning, but we matched the bin count for a fair comparison).

- **Positional Embedding**: Uses four types of absolute positional embeddings, which are summed: (1) Learned PE based on the hospital visit number, (2) Learned PE based on token type, (3) Time embedding based on age using the Time2Vec model (Kazemi et al., 2019), and (4) Position-based sin/cos positional embedding from Vaswani et al. (2017).

- **Learning Objective**: Uses a next token prediction loss.

**ETHOS**   (Total Parameters: 20,050,944)

- **Backbone**: Transformer decoder.

- **Bin Sharing**: Yes, Feature and Value are tokenized separately. Numeric values share bin tokens across different features.

- **Binning**: 10-uniform binning.

- **Time Information**: Following the original paper, the time interval between each event is converted into one of 13 discrete tokens, which are inserted into the sequence between event tokens.

- **Positional Embedding**: Uses a learned positional embedding based on position.

- **Learning Objective**: Uses NTP loss.

### D.3 Pre-training Details

Pre-training was conducted using the training dataset. The hyperparameters were fixed as follows: the batch size was 64, and the number of training epochs was 50. We used the Adam (Adam et al., 2014) optimizer with a learning rates of $\{5\times10^{-4}, 1\times10^{-4}, 5\times10^{-5}, 1\times10^{-5}\}$. A 50-step warmup was employed, followed by a cosine annealing schedule that reduced the learning rate to 1/100 of its initial value. Gradient clipping was applied with a threshold of 1.0. The training was performed using either 4 NVIDIA A40 GPUs or 2 NVIDIA RTX PRO 6000 Blackwell GPUs, with Distributed Data Parallel training at a per-GPU batch size of 16 or 32, respectively. The implementation was based on Python version 3.12 and PyTorch (Paszke et al., 2017) version 2.8.0.

## E Downstream tasks and Fine-tuning

### E.1 Downstream tasks

Existing EHR foundation models generally lack generative capabilities, and thus evaluating performance on diverse clinical downstream tasks has been a common practice (Fallahpour et al., 2025; Huang et al., 2024; Burkhart et al., 2025; Renc et al., 2024). While our model possesses strong generative properties, we follow the line of prior work and perform downstream task evaluations to assess the quality of patient representations at each timestamp. We first adopted the four widely used downstream tasks from the MIMIC-III benchmark (Harutyunyan et al., 2019): In-hospital Mortality (IHM), Decompensation-death (Dec-D), Length of Stay (LOS), and Phenotyping (Phe). For these tasks, labels were obtained following the original preprocessing pipelines. To further examine whether the model can capture patient states beyond simple deterioration, we added three additional tasks: Decompensation-arrest (Dec-A), Oliguria/Anuria (HUO), and Vasopressor (Vaso) use. Dec-A is similar to Dec-D, but includes arrest events from CHARTEVENTS.csv in addition to death as decompensation events; the task aims to predict deterioration 24 hours in advance. HUO labels indicate whether the patient currently exhibits oliguria or anuria. Specifically, oliguria is defined as urine output below 0.5 mL/kg/hr for at least 6 hours, and anuria as below 0.1 mL/kg/hr for at least 6 hours. Labels were set for patients with available weight and hourly urine output data, while patients with incomplete information were excluded from training. Vaso labels indicate whether a patient is currently receiving vasopressors. Positive labels were assigned if administration records for Vasopressin, Dobutamine, Epinephrine, Norepinephrine, or Dopamine were present in INPUTEVENTS_CV.csv or INPUTEVENTS_MV.csv; negative labels were assigned if other medications were administered but no vasopressors were recorded.

All downstream tasks were measured at the hospitalization level following the original papers (Harutyunyan et al., 2019) (i.e., if a patient had multiple hospital admissions, each admission was treated as a separate EHR sequence). Table 8 provides detailed statistics, including the number of hospitalizations available for each task and the label distribution for each class.

### E.2 Fine-tuning details

The downstream tasks were trained by attaching a task-specific prediction head to the last hidden state of the backbone, applied uniformly to our model and all baselines. Since labels must be inferred using only information available up to each timestep, a causal mask was employed. We froze the pretrained model and trained the prediction heads simultaneously. To evaluate generalization performance under varying input distributions (e.g., absence of laboratory data), we experimented with three settings: (i) the full set of 117 variables (17 vital signs + 100 laboratory measurements), (ii) only 17 vital signs, and (iii) a reduced set of 6 vital signs (SBP, DBP, body temperature, heart rate, respiratory rate, SpO$_2$). For training, the original training set was split into train/validation subsets with an 85:15 ratio; the train subset was used for optimization, while the validation subset was used for early stopping and hyperparameter search. As in pre-training, the learning rate $lr$ was selected from $5 \times 10^{-4}, 1 \times 10^{-4}, 5 \times 10^{-5}, 1 \times 10^{-5}$. The batch size was fixed 100. Training was performed for 5 epochs with a 50-step warm-up followed by cosine annealing that decayed the learning rate to $1/100$ of its initial value.

Each task-specific prediction head consisted of a two-layer MLP. To account for label imbalance, task losses were weighted according to label frequencies in the training dataset. For efficiency,

Table 8: Prediction time and label counts for each downstream task. For binary classification, the positive/negative counts are shown; for multiclass classification, the counts for each class are shown. For tasks with multiple sub-tasks, each sub-task's label counts are shown.

| Task | Prediction point | # Hospitalization | # Sub task & # Class | Train labels | Test labels |
|------|------------------|-------------------|----------------------|--------------|-------------|
| IHM | 48h after ICU adm. | 17,903 / 3,236 | 1 / 2 | 2424 / 15479 | 374 / 2,862 |
| Dec-D | Hourly | 35,365 / 6,237 | 1 / 2 | 61,018 / 2,847,424 | 9,684 / 513,552 |
| Dec-A | Hourly | 35,365 / 6,237 | 1 / 2 | 65,298 / 2,843,144 | 10,239 / 512,997 |
| LOS | Hourly | 35,523 / 6,265 | 1 / 10 | [cls1: 790,196, cls2: 503,423, cls3: 316,774, cls4: 215,075, cls5: 158,987, cls6: 124,146, cls7: 100,890, cls8: 84,241, cls9: 312,111, cls10: 319,619] | [cls1: 139,682, cls2: 90,478, cls3: 56,289, cls4: 38,795, cls5: 28,542, cls6: 22,225, cls7: 18,077, cls8: 15,145, cls9: 55,997, cls10: 60,710] |
| Phe | End of stay | 35,563 / 6,273 | 25 / 2 | t1: 7,644 / 28,029
t2: 2660 / 33013
t3: 3657 / 32016
t4: 11434 / 24239
t5: 4821 / 30852
t6: 4675 / 30998
t7: 7349 / 28324
t8: 2565 / 33108
t9: 9550 / 26123
t10: 11497 / 24176
t11: 3444 / 32229
t12: 6869 / 28804
t13: 10362 / 25311
t14: 14922 / 20751
t15: 9617 / 26056
t16: 2573 / 33100
t17: 4775 / 30898
t18: 3170 / 32503
t19: 1816 / 33857
t20: 1435 / 34238
t21: 3080 / 32593
t22: 4970 / 30703
t23: 6468 / 29205
t24: 5118 / 30555
t25: 2779 / 32894 | t1: 374 / 2862
t2: 1331 / 4945
t3: 415 / 5861
t4: 675 / 5601
t5: 2028 / 4248
t6: 831 / 5445
t7: 789 / 5487
t8: 1337 / 4939
t9: 442 / 5834
t10: 1683 / 4593
t11: 2074 / 4202
t12: 593 / 5683
t13: 1205 / 5071
t14: 1813 / 4463
t15: 2653 / 3623
t16: 1667 / 4609
t17: 495 / 5781
t18: 819 / 5457
t19: 556 / 5720
t20: 355 / 5921
t21: 272 / 6004
t22: 570 / 5706
t23: 852 / 5424
t24: 1111 / 5165
t25: 874 / 5402 |
| HUO | Hourly | 11,891 / 2,187 | 2 / 2 | t1: 148737 / 891153
t2: 148737 / 1053830 | t1: 56102 / 162429
t2: 26682 / 191849 |
| Vaso | Hourly | 35,438 / 6,249 | 1 / 2 | 202,765 / 2,747,736 | 36,306 / 494,215 |

downstream tasks were conducted in a multi-task setting where all seven tasks were jointly optimized; the final objective was defined as the average of the task-specific losses. Since probing does not allow training of new tokens, we instead introduced a `<Birth>` token at each timestep, serving the same role as the `<SOS>` token, and used its representation for task prediction. For baseline models, we preserved their original binning and embedding procedures, while unifying the learning objective to the downstream tasks. All downstream tasks were conducted on a single NVIDIA A40 GPU.

# F RESULTS

## F.1 DOWNSTREAM TASK RESULTS

Tables 9, 10, and 11 report the results of all downstream tasks. Table 9 presents the loss, number of features used, and selected pretraining/downstream-task learning rate for each experiment, while the remaining results are shown in Tables 10 and 11.

## F.2 EHR GENERATION AND EVALUATION

We generated EHRs with a low temperature of 0.2. Below are examples generated from the same initial EHR, shown in order for our model (Figure 8) and the ETHOS model (Figure 9).

Figure 10 shows the prompt we used for LLM-based evaluation.

Table 9: Meta Information

| Method | Value sharing | Downstream # features | test-loss | valid-loss | train-loss | Pre-training LR | Downstream LR |
|---|---|---|---|---|---|---|---|
| HEART | X | 117 | 5.3043 | 5.3681 | 5.3714 | 0.0005 | 0.0005 |
| HEART | X | 17 | 5.4340 | 5.4604 | 5.4713 | 0.0005 | 0.0005 |
| HEART | X | 6 | 5.8346 | 5.8703 | 5.8702 | 0.0005 | 0.0005 |
| MOTOR | X | 117 | 4.9454 | 4.9020 | 4.9118 | 0.0001 | 0.0005 |
| MOTOR | X | 17 | 5.2117 | 5.1570 | 5.1803 | 0.0001 | 0.0005 |
| MOTOR | X | 6 | 5.6451 | 5.6455 | 5.6519 | 0.0001 | 0.0005 |
| EHRSHOT | X | 117 | 5.8406 | 5.9138 | 5.9241 | 0.0005 | 0.0005 |
| EHRSHOT | X | 17 | 6.0777 | 6.1018 | 6.1022 | 0.0005 | 0.0005 |
| EHRSHOT | X | 6 | 6.3406 | 6.3509 | 6.3325 | 0.0005 | 0.0005 |
| TRADE | X | 117 | 5.2599 | 5.2480 | 5.2807 | 0.0005 | 0.0005 |
| TRADE | X | 17 | 5.4538 | 5.4529 | 5.4839 | 0.0005 | 0.0005 |
| TRADE | X | 6 | 6.0481 | 6.0643 | 6.0702 | 0.0005 | 0.0005 |
| EHRmamba | X | 117 | 5.1366 | 5.1603 | 5.2125 | 0.0005 | 0.0005 |
| EHRmamba | X | 17 | 5.4389 | 5.4228 | 5.4678 | 0.0005 | 0.0005 |
| EHRmamba | X | 6 | 5.9261 | 5.9404 | 5.9573 | 0.0005 | 0.0005 |
| Ours (No value share) | X | 117 | 4.6861 | 4.6386 | 4.6273 | 0.0005 | 0.0005 |
| Ours (No value share) | X | 17 | 4.9070 | 4.8621 | 4.8664 | 0.0005 | 0.0005 |
| Ours (No value share) | X | 6 | 5.3675 | 5.3538 | 5.4045 | 0.0005 | 0.0005 |
| FM4EHR | O | 117 | 6.4288 | 6.4611 | 6.4344 | 0.0005 | 0.0001 |
| FM4EHR | O | 17 | 6.3888 | 6.4391 | 6.3969 | 0.0005 | 0.0005 |
| FM4EHR | O | 6 | 6.3972 | 6.4394 | 6.4069 | 0.0005 | 0.0005 |
| ETHOS | O | 117 | 4.9710 | 5.0374 | 4.9378 | 0.0005 | 0.0001 |
| ETHOS | O | 17 | 5.2479 | 5.2044 | 5.1124 | 0.0005 | 0.0001 |
| ETHOS | O | 6 | 5.5724 | 5.5714 | 5.5213 | 0.0005 | 0.0001 |
| STraTS | O | 117 | 5.7857 | 5.8492 | 5.8505 | 0.0001 | 0.0005 |
| STraTS | O | 17 | 5.8123 | 5.8335 | 5.8446 | 0.0001 | 0.0005 |
| STraTS | O | 6 | 6.0711 | 6.0760 | 6.0834 | 0.0001 | 0.0005 |
| Ours (Value share) | O | 117 | 4.8786 | 4.8954 | 5.0276 | 0.0005 | 0.0005 |
| Ours (Value share) | O | 117 | 5.0430 | 5.0862 | 5.1887 | 0.0005 | 0.0001 |
| Ours (Value share) | O | 6 | 5.5613 | 5.6554 | 5.7523 | 0.0005 | 0.0005 |

Table 10: Performance Results (Part 1); I., and P. denote In-Hospital-Mortality, and Phenotyping respectively. R., and P. denote AUROC and AUPRC respectively.

| Methods | Value sharing | L-R | L-P | P-R@0 | P-R0 | P-R1 | P-P1 | P-R2 | P-P2 | P-R3 | P-P3 | P-R4 | P-P4 | P-R5 | P-P5 | P-R6 | P-P6 | P-R7 | P-P7 | P-R8 | P-P8 | P-R9 | P-P9 | P-R10 | P-P10 | P-R11 | P-P11 | P-R12 | P-P12 | P-R13 | P-P13 | P-R14 | P-P14 | P-R15 | P-P15 | P-R16 | P-P16 |
|---|---|---|---|---|---|---|---|---|---|---|---|---|---|---|---|---|---|---|---|---|---|---|---|---|---|---|---|---|---|---|---|---|---|---|---|---|---|
| HEART | ✗ | 0.8379 | 0.4423 | 0.7784 | 0.4668 | 0.7973 | 0.2444 | 0.7143 | 0.2230 | 0.6313 | 0.4195 | 0.7270 | 0.2590 | 0.6785 | 0.2150 | 0.6174 | 0.2753 | 0.7025 | 0.1677 | 0.7211 | 0.4408 | 0.7417 | 0.5820 | 0.8353 | 0.4154 | 0.7119 | 0.3330 | 0.6655 | 0.4657 | 0.6064 | 0.5210 | 0.6996 | 0.4322 | 0.6689 | 0.1210 | 0.7242 | 0.2543 |
| HEART | ✗ | 0.8198 | 0.3989 | 0.7179 | 0.3948 | 0.7932 | 0.2283 | 0.6423 | 0.1737 | 0.6187 | 0.4145 | 0.6870 | 0.2286 | 0.6679 | 0.2005 | 0.6167 | 0.2811 | 0.6886 | 0.1545 | 0.6910 | 0.4164 | 0.7209 | 0.5593 | 0.8415 | 0.4266 | 0.7287 | 0.3465 | 0.6681 | 0.4397 | 0.5826 | 0.4917 | 0.6679 | 0.4001 | 0.6260 | 0.1110 | 0.6856 | 0.2256 |
| HEART | ✗ | 0.7391 | 0.2839 | 0.6813 | 0.3525 | 0.7061 | 0.1494 | 0.6335 | 0.1574 | 0.6168 | 0.4019 | 0.6091 | 0.1798 | 0.6700 | 0.2113 | 0.5934 | 0.2550 | 0.6925 | 0.1342 | 0.6895 | 0.4207 | 0.6994 | 0.4991 | 0.6117 | 0.1442 | 0.6490 | 0.2405 | 0.7100 | 0.4029 | 0.5087 | 0.5125 | 0.6302 | 0.3732 | 0.5593 | 0.0971 | 0.6513 | 0.1899 |
| MOTOR | ✗ | 0.8724 | 0.5465 | 0.8392 | 0.5832 | 0.8732 | 0.4259 | 0.7342 | 0.2799 | 0.7162 | 0.5116 | 0.8596 | 0.5353 | 0.7506 | 0.3126 | 0.6648 | 0.3434 | 0.7426 | 0.1765 | 0.7920 | 0.4966 | 0.7938 | 0.6587 | 0.7566 | 0.3195 | 0.6580 | 0.2888 | 0.7233 | 0.5048 | 0.7100 | 0.6214 | 0.7354 | 0.4857 | 0.7443 | 0.1887 | 0.8503 | 0.4827 |
| MOTOR | ✗ | 0.8406 | 0.4612 | 0.7440 | 0.4240 | 0.8743 | 0.4176 | 0.6969 | 0.2043 | 0.6940 | 0.4910 | 0.7501 | 0.3115 | 0.7293 | 0.2733 | 0.3287 | 0.3287 | 0.7311 | 0.1707 | 0.7492 | 0.3768 | 0.7812 | 0.6392 | 0.7575 | 0.3195 | 0.6217 | 0.2587 | 0.7053 | 0.4761 | 0.6762 | 0.5788 | 0.6899 | 0.4337 | 0.6892 | 0.1472 | 0.7348 | 0.2838 |
| MOTOR | ✗ | 0.7611 | 0.3396 | 0.7065 | 0.3770 | 0.7528 | 0.2513 | 0.6739 | 0.1951 | 0.7000 | 0.5000 | 0.6871 | 0.2348 | 0.7272 | 0.2644 | 0.2634 | 0.2634 | 0.6371 | 0.1085 | 0.7313 | 0.2388 | 0.7520 | 0.5708 | 0.6449 | 0.1302 | 0.4777 | 0.1859 | 0.6657 | 0.4421 | 0.6657 | 0.5589 | 0.6568 | 0.3877 | 0.6066 | 0.1216 | 0.6915 | 0.2329 |
| EHRSHOT | ✗ | 0.8010 | 0.4327 | 0.7440 | 0.4052 | 0.5883 | 0.0898 | 0.5557 | 0.1296 | 0.4747 | 0.3078 | 0.6799 | 0.2224 | 0.6318 | 0.1713 | 0.5884 | 0.2014 | 0.6201 | 0.0606 | 0.6590 | 0.2611 | 0.5844 | 0.4074 | 0.6201 | 0.0836 | 0.4716 | 0.1758 | 0.6032 | 0.3820 | 0.5812 | 0.4857 | 0.6789 | 0.3945 | 0.6307 | 0.1164 | 0.6681 | 0.2162 |
| EHRSHOT | ✗ | 0.7199 | 0.2864 | 0.5946 | 0.2861 | 0.5461 | 0.0757 | 0.5534 | 0.1248 | 0.4827 | 0.2753 | 0.5724 | 0.1710 | 0.5418 | 0.1386 | 0.4724 | 0.1959 | 0.4329 | 0.1012 | 0.4522 | 0.2388 | 0.4971 | 0.3246 | 0.4492 | 0.0829 | 0.3585 | 0.2198 | 0.5202 | 0.3105 | 0.4923 | 0.4161 | 0.5581 | 0.2479 | 0.5458 | 0.0887 | 0.5575 | 0.1566 |
| EHRSHOT | ✗ | 0.6235 | 0.1835 | 0.6428 | 0.2999 | 0.5890 | 0.0688 | 0.7246 | 0.1318 | 0.4203 | 0.2753 | 0.6172 | 0.1798 | 0.6161 | 0.1692 | 0.4635 | 0.3333 | 0.6201 | 0.1543 | 0.4932 | 0.5430 | 0.3543 | 0.3543 | 0.4485 | 0.0829 | 0.5395 | 0.2784 | 0.5395 | 0.3195 | 0.5158 | 0.4290 | 0.5323 | 0.3043 | 0.4427 | 0.0681 | 0.6012 | 0.1767 |
| TRADE | ✗ | 0.8276 | 0.4406 | 0.8314 | 0.3590 | 0.8106 | 0.2613 | 0.7246 | 0.2258 | 0.4851 | 0.4851 | 0.8236 | 0.4264 | 0.7015 | 0.2542 | 0.6618 | 0.3333 | 0.7102 | 0.1543 | 0.7762 | 0.5430 | 0.7747 | 0.6194 | 0.7692 | 0.2961 | 0.6473 | 0.2784 | 0.6941 | 0.4793 | 0.6206 | 0.5323 | 0.7379 | 0.4886 | 0.7143 | 0.1669 | 0.8214 | 0.4197 |
| TRADE | ✗ | 0.8054 | 0.3719 | 0.6876 | 0.3816 | 0.8205 | 0.2722 | 0.6679 | 0.1766 | 0.4566 | 0.4566 | 0.6806 | 0.2307 | 0.6382 | 0.1967 | 0.3307 | 0.3067 | 0.6355 | 0.1226 | 0.6980 | 0.4338 | 0.6961 | 0.5029 | 0.7306 | 0.2442 | 0.6401 | 0.2876 | 0.6620 | 0.4350 | 0.5491 | 0.4646 | 0.6474 | 0.4033 | 0.3829 | 0.0998 | 0.6394 | 0.1983 |
| TRADE | ✗ | 0.6726 | 0.2184 | 0.6533 | 0.3408 | 0.6695 | 0.1498 | 0.6070 | 0.1502 | 0.6373 | 0.4520 | 0.6032 | 0.1858 | 0.6141 | 0.1826 | 0.2887 | 0.2887 | 0.6280 | 0.1152 | 0.6652 | 0.4180 | 0.6647 | 0.4574 | 0.5914 | 0.1332 | 0.5835 | 0.2109 | 0.5861 | 0.3660 | 0.6041 | 0.5224 | 0.6034 | 0.3469 | 0.5538 | 0.1002 | 0.5995 | 0.1757 |
| EHRmamba | ✗ | 0.8677 | 0.5565 | 0.7765 | 0.5018 | 0.7149 | 0.1678 | 0.6756 | 0.3126 | 0.6447 | 0.4364 | 0.7657 | 0.2964 | 0.6304 | 0.1736 | 0.5904 | 0.2567 | 0.6865 | 0.1406 | 0.7236 | 0.4395 | 0.7457 | 0.6117 | 0.7457 | 0.1330 | 0.6077 | 0.2467 | 0.6345 | 0.4229 | 0.5554 | 0.5919 | 0.6973 | 0.4410 | 0.6709 | 0.1419 | 0.7462 | 0.2658 |
| EHRmamba | ✗ | 0.8125 | 0.4016 | 0.6265 | 0.3172 | 0.7063 | 0.1601 | 0.6672 | 0.1769 | 0.6328 | 0.4436 | 0.6588 | 0.2140 | 0.5783 | 0.1578 | 0.6149 | 0.6149 | 0.6661 | 0.1245 | 0.6319 | 0.3601 | 0.6948 | 0.5227 | 0.5642 | 0.1347 | 0.5962 | 0.2386 | 0.6663 | 0.4479 | 0.4935 | 0.4648 | 0.6194 | 0.3713 | 0.5060 | 0.0874 | 0.6245 | 0.1906 |
| EHRmamba | ✗ | 0.7048 | 0.3109 | 0.6112 | 0.2996 | 0.6305 | 0.1175 | 0.5695 | 0.1370 | 0.5785 | 0.3952 | 0.5616 | 0.1582 | 0.5843 | 0.1593 | 0.5997 | 0.2835 | 0.6672 | 0.1240 | 0.6283 | 0.3696 | 0.7142 | 0.5345 | 0.5683 | 0.1266 | 0.5617 | 0.2259 | 0.5562 | 0.3363 | 0.4181 | 0.4181 | 0.5733 | 0.3232 | 0.5272 | 0.0907 | 0.5794 | 0.1624 |
| Ours (No value share) | ✗ | 0.8891 | 0.6009 | 0.8763 | 0.6506 | 0.8992 | 0.4789 | 0.8412 | 0.5324 | 0.9938 | 0.5938 | 0.9041 | 0.6281 | 0.7616 | 0.3442 | 0.6994 | 0.3482 | 0.7902 | 0.2504 | 0.8273 | 0.6372 | 0.8321 | 0.7077 | 0.8711 | 0.4865 | 0.7581 | 0.4072 | 0.7426 | 0.5281 | 0.7292 | 0.6327 | 0.7648 | 0.5438 | 0.8219 | 0.3090 | 0.8954 | 0.9912 |
| Ours (No value share) | ✗ | 0.8694 | 0.5446 | 0.7552 | 0.4393 | 0.8902 | 0.4623 | 0.7563 | 0.2751 | 0.5311 | 0.5311 | 0.7828 | 0.3644 | 0.7305 | 0.2812 | 0.6859 | 0.3440 | 0.7652 | 0.2255 | 0.7689 | 0.5301 | 0.8057 | 0.6630 | 0.8538 | 0.4509 | 0.7600 | 0.4059 | 0.7271 | 0.5027 | 0.6883 | 0.5842 | 0.7133 | 0.4612 | 0.6922 | 0.1618 | 0.7618 | 0.3246 |
| Ours (No value share) | ✗ | 0.8012 | 0.4016 | 0.7237 | 0.4052 | 0.7655 | 0.2476 | 0.7265 | 0.2367 | 0.5348 | 0.5348 | 0.7229 | 0.2800 | 0.7274 | 0.2751 | 0.6562 | 0.3181 | 0.7596 | 0.2041 | 0.7515 | 0.5038 | 0.7840 | 0.6167 | 0.6887 | 0.2057 | 0.6263 | 0.2499 | 0.6956 | 0.4575 | 0.6808 | 0.5771 | 0.6749 | 0.4049 | 0.6170 | 0.1203 | 0.7228 | 0.2589 |
| FM4EHR | ○ | 0.6172 | 0.1770 | 0.4513 | 0.1876 | 0.5803 | 0.0886 | 0.4661 | 0.0980 | 0.4534 | 0.2972 | 0.5673 | 0.1619 | 0.5155 | 0.1266 | 0.5571 | 0.2488 | 0.4186 | 0.0567 | 0.5245 | 0.2789 | 0.5675 | 0.3785 | 0.5384 | 0.1065 | 0.5043 | 0.1923 | 0.5176 | 0.2966 | 0.5315 | 0.4459 | 0.5974 | 0.3251 | 0.4676 | 0.0735 | 0.5518 | 0.1572 |
| FM4EHR | ○ | 0.5773 | 0.1519 | 0.5000 | 0.2121 | 0.5000 | 0.0661 | 0.5000 | 0.1076 | 0.5000 | 0.3231 | 0.5000 | 0.1324 | 0.5000 | 0.1257 | 0.5000 | 0.2130 | 0.5000 | 0.0704 | 0.5000 | 0.2682 | 0.5000 | 0.3305 | 0.5000 | 0.0945 | 0.5000 | 0.1920 | 0.5000 | 0.2889 | 0.5000 | 0.4227 | 0.5000 | 0.2656 | 0.5000 | 0.0789 | 0.5000 | 0.1305 |
| FM4EHR | ○ | 0.4965 | 0.1150 | 0.5000 | 0.2121 | 0.5000 | 0.0661 | 0.5000 | 0.1076 | 0.5000 | 0.3231 | 0.5000 | 0.1324 | 0.5000 | 0.1257 | 0.5000 | 0.2130 | 0.5000 | 0.0704 | 0.5000 | 0.2682 | 0.5000 | 0.3305 | 0.5000 | 0.0945 | 0.5000 | 0.1920 | 0.5000 | 0.2889 | 0.5000 | 0.4227 | 0.5000 | 0.2656 | 0.5000 | 0.0789 | 0.5000 | 0.1305 |
| ETHOS | ○ | 0.8586 | 0.3501 | 0.7833 | 0.4778 | 0.8609 | 0.3690 | 0.9425 | 0.2447 | 0.6984 | 0.4948 | 0.7735 | 0.3581 | 0.6873 | 0.2044 | 0.6591 | 0.3335 | 0.7387 | 0.1715 | 0.7580 | 0.5135 | 0.7786 | 0.6239 | 0.8043 | 0.3517 | 0.6861 | 0.3039 | 0.7054 | 0.4710 | 0.6481 | 0.5556 | 0.7006 | 0.4431 | 0.7466 | 0.2565 | 0.7667 | 0.3125 |
| ETHOS | ○ | 0.8173 | 0.3122 | 0.7222 | 0.3959 | 0.8499 | 0.3005 | 0.5000 | 0.1775 | 0.6914 | 0.4720 | 0.7233 | 0.2830 | 0.6735 | 0.2016 | 0.6535 | 0.3293 | 0.7383 | 0.1721 | 0.7432 | 0.4919 | 0.7721 | 0.6178 | 0.7256 | 0.3158 | 0.6726 | 0.3013 | 0.7011 | 0.4855 | 0.6587 | 0.5397 | 0.6732 | 0.4109 | 0.6696 | 0.1173 | 0.6967 | 0.2395 |
| ETHOS | ○ | 0.7558 | 0.6759 | 0.6759 | 0.3273 | 0.7145 | 0.1814 | 0.5000 | 0.1708 | 0.6936 | 0.4632 | 0.6849 | 0.2426 | 0.5869 | 0.1947 | 0.6385 | 0.3195 | 0.7456 | 0.1507 | 0.4579 | 0.4579 | 0.5695 | 0.5695 | 0.6101 | 0.1401 | 0.6048 | 0.2280 | 0.6422 | 0.3686 | 0.6501 | 0.5564 | 0.6348 | 0.3605 | 0.5881 | 0.1015 | 0.6127 | 0.1779 |
| STraTS | ○ | 0.7592 | 0.3111 | 0.6583 | 0.3269 | 0.7350 | 0.1876 | 0.6662 | 0.1775 | 0.6887 | 0.4720 | 0.6873 | 0.2456 | 0.6735 | 0.1852 | 0.5766 | 0.2465 | 0.7456 | 0.1617 | 0.7300 | 0.4761 | 0.7466 | 0.5275 | 0.6101 | 0.1470 | 0.6048 | 0.2384 | 0.6422 | 0.3686 | 0.6135 | 0.4966 | 0.5993 | 0.3242 | 0.5994 | 0.1081 | 0.6735 | 0.2206 |
| STraTS | ○ | 0.7062 | 0.2249 | 0.6874 | 0.3455 | 0.7805 | 0.2233 | 0.6713 | 0.1708 | 0.6863 | 0.4695 | 0.7977 | 0.2322 | 0.6739 | 0.1825 | 0.5939 | 0.2614 | 0.7399 | 0.1606 | 0.7313 | 0.4765 | 0.7425 | 0.5236 | 0.6634 | 0.1976 | 0.6602 | 0.2373 | 0.6419 | 0.3734 | 0.6227 | 0.5020 | 0.6281 | 0.3470 | 0.6133 | 0.1114 | 0.6679 | 0.2239 |
| STraTS | ○ | 0.6993 | 0.2192 | 0.6472 | 0.3141 | 0.6621 | 0.1421 | 0.6552 | 0.1708 | 0.6863 | 0.4632 | 0.7126 | 0.2556 | 0.6747 | 0.1803 | 0.5562 | 0.2477 | 0.7445 | 0.1654 | 0.7213 | 0.4679 | 0.7430 | 0.5289 | 0.6042 | 0.1459 | 0.5940 | 0.2299 | 0.6474 | 0.3715 | 0.6217 | 0.5064 | 0.6084 | 0.3324 | 0.5547 | 0.0969 | 0.6636 | 0.2087 |
| Ours (Value share) | ○ | 0.8760 | 0.5586 | 0.8540 | 0.5745 | 0.8942 | 0.4188 | 0.8394 | 0.5054 | 0.7162 | 0.5124 | 0.8512 | 0.4959 | 0.7152 | 0.2714 | 0.6800 | 0.3453 | 0.7572 | 0.1885 | 0.8091 | 0.6081 | 0.8279 | 0.7040 | 0.8403 | 0.4538 | 0.6886 | 0.3019 | 0.7278 | 0.9097 | 0.6678 | 0.5782 | 0.7534 | 0.4967 | 0.7677 | 0.1927 | 0.8345 | 0.4477 |
| Ours (Value share) | ○ | 0.8649 | 0.5191 | 0.8297 | 0.5470 | 0.8603 | 0.3063 | 0.7747 | 0.4233 | 0.6637 | 0.4540 | 0.7977 | 0.3551 | 0.6362 | 0.1834 | 0.6466 | 0.2999 | 0.7072 | 0.1478 | 0.7331 | 0.4609 | 0.7776 | 0.6534 | 0.7502 | 0.2489 | 0.6360 | 0.2687 | 0.6762 | 0.4728 | 0.6171 | 0.5319 | 0.7419 | 0.4725 | 0.7141 | 0.1509 | 0.7680 | 0.3210 |
| Ours (Value share) | ○ | 0.7751 | 0.3660 | 0.7033 | 0.3846 | 0.7115 | 0.1505 | 0.6860 | 0.1810 | 0.7075 | 0.5073 | 0.7126 | 0.2556 | 0.7096 | 0.2347 | 0.6440 | 0.3138 | 0.7348 | 0.1635 | 0.7359 | 0.4696 | 0.7731 | 0.6095 | 0.6084 | 0.1393 | 0.6019 | 0.2362 | 0.7043 | 0.4712 | 0.6479 | 0.5555 | 0.6754 | 0.4082 | 0.5690 | 0.0979 | 0.6799 | 0.2386 |

Table 11: Performance Results (Part 2); P., L., H., and V. denote Phenotyping, Length-of-Stay, Oliguria/Anuria, and Vasopressor respectively. R. and P. denote AUROC and AUPRC respectively.

| Method | Value sharing | P.R.17 | P.P.17 | P.R.18 | P.P.18 | P.R.19 | P.P.19 | P.R.20 | P.P.20 | P.R.21 | P.R.22 | P.P.22 | P.R.23 | P.P.23 | P.R.24 | P.P.24 | P.R.25 | P.P.25 | P.R.26 | P.P.26 | P.R.27 | P.P.27 | P.R.28 | P.P.28 | P.R.29 | P.P.29 | L.kappa | L.logloss | H.R.0 | H.P.0 | H.R.1 | H.P.1 | H.macro R. | H.micro R. | V.R. | V.P. |
|---|---|---|---|---|---|---|---|---|---|---|---|---|---|---|---|---|---|---|---|---|---|---|---|---|---|---|---|---|---|---|---|---|---|---|---|---|
| HEART | X | 0.7663 | 0.2522 | 0.6253 | 0.0866 | 0.6792 | 0.0899 | 0.6448 | 0.1384 | 0.1774 | 0.5902 | 0.1152 | 0.6797 | 0.2104 | 0.6436 | 0.1797 | 0.6245 | 0.0708 | 0.6552 | 0.2559 | 0.6649 | 0.2723 | 0.6749 | 0.1229 | 0.6304 | 0.1821 | 0.1417 | 2.1418 | 0.6676 | 0.4354 | 0.7383 | 0.3380 | 0.7029 | 0.7014 | 0.8652 | 0.3629 |
| HEART | X | 0.6654 | 0.1601 | 0.6148 | 0.0833 | 0.6478 | 0.0780 | 0.6340 | 0.1311 | 0.1755 | 0.6004 | 0.1110 | 0.6591 | 0.2012 | 0.6356 | 0.1804 | 0.6099 | 0.0690 | 0.6339 | 0.2499 | 0.6468 | 0.2610 | 0.6357 | 0.1102 | 0.6258 | 0.1789 | 0.1361 | 2.1136 | 0.6274 | 0.3772 | 0.6634 | 0.2519 | 0.6454 | 0.6558 | 0.8554 | 0.3343 |
| HEART | X | 0.6442 | 0.1424 | 0.6047 | 0.0877 | 0.6094 | 0.0693 | 0.6339 | 0.1370 | 0.1668 | 0.5838 | 0.1071 | 0.6410 | 0.1923 | 0.6168 | 0.1751 | 0.6057 | 0.0665 | 0.6326 | 0.2447 | 0.6397 | 0.2546 | 0.6242 | 0.1087 | 0.6069 | 0.1740 | 0.1035 | 2.2318 | 0.7112 | 0.3637 | 0.6518 | 0.2360 | 0.6349 | 0.6372 | 0.8211 | 0.2529 |
| MOTOR | X | 0.8438 | 0.4490 | 0.6725 | 0.1139 | 0.7483 | 0.1088 | 0.6944 | 0.1732 | 0.2179 | 0.6282 | 0.1392 | 0.7252 | 0.2442 | 0.6645 | 0.2050 | 0.6442 | 0.0810 | 0.6998 | 0.2911 | 0.7268 | 0.3037 | 0.6783 | 0.1417 | 0.6773 | 0.2059 | 0.1565 | 2.0410 | 0.6574 | 0.5027 | 0.7942 | 0.2952 | 0.7527 | 0.7483 | 0.8905 | 0.4381 |
| MOTOR | X | 0.7263 | 0.2287 | 0.6655 | 0.1123 | 0.7036 | 0.0998 | 0.6818 | 0.1638 | 0.2065 | 0.6192 | 0.1331 | 0.6947 | 0.2258 | 0.6506 | 0.1991 | 0.6639 | 0.0775 | 0.6873 | 0.2820 | 0.7014 | 0.2946 | 0.6677 | 0.1374 | 0.6510 | 0.1975 | 0.1426 | 2.0695 | 0.6533 | 0.4456 | 0.7006 | 0.4343 | 0.6790 | 0.6869 | 0.8708 | 0.3958 |
| MOTOR | X | 0.6971 | 0.2001 | 0.6592 | 0.1128 | 0.6720 | 0.0915 | 0.6552 | 0.1573 | 0.2006 | 0.6064 | 0.1079 | 0.6725 | 0.1952 | 0.6376 | 0.1919 | 0.6364 | 0.0751 | 0.6684 | 0.2707 | 0.6779 | 0.2828 | 0.6441 | 0.1338 | 0.6313 | 0.1890 | 0.1389 | 2.1351 | 0.6159 | 0.4472 | 0.6996 | 0.2989 | 0.6803 | 0.6809 | 0.8327 | 0.3066 |
| EHRSHOT | X | 0.7122 | 0.2296 | 0.5745 | 0.0907 | 0.5941 | 0.0890 | 0.5773 | 0.1268 | 0.1623 | 0.5437 | 0.1045 | 0.6020 | 0.1852 | 0.5702 | 0.1636 | 0.5570 | 0.0638 | 0.5886 | 0.2465 | 0.5948 | 0.2580 | 0.5537 | 0.1076 | 0.5700 | 0.1690 | 0.1208 | 2.1553 | 0.5727 | 0.3916 | 0.6764 | 0.2674 | 0.6518 | 0.6487 | 0.8554 | 0.3469 |
| EHRSHOT | X | 0.6366 | 0.1581 | 0.5568 | 0.0856 | 0.5626 | 0.0844 | 0.5650(?) | 0.1241 | 0.1572 | 0.5145 | 0.0992 | 0.5743 | 0.1792 | 0.5442 | 0.1565 | 0.5399 | 0.0610 | 0.5620 | 0.2392 | 0.5666 | 0.2485 | 0.5382 | 0.1045 | 0.5408 | 0.1596 | 0.1004 | 2.1936 | 0.5459 | 0.3542 | 0.6262 | 0.2369 | 0.6177 | 0.6132 | 0.7844 | 0.2464 |
| EHRSHOT | X | 0.6190 | 0.1403 | 0.5340 | 0.0807 | 0.5340 | 0.0803 | 0.5773 | 0.1694 | 0.2128 | 0.5402 | 0.1003(?) | 0.6020 | 0.2406 | 0.5702 | 0.2017 | 0.5570 | 0.0797 | 0.5886 | 0.2877 | 0.5948 | 0.3006 | 0.5537 | 0.1399 | 0.5700 | 0.2031 | 0.1673 | 2.2127 | 0.7118 | 0.5069 | 0.7860 | 0.4370 | 0.7552 | 0.7495 | 0.7415 | 0.2041 |
| TRADE | X | 0.8352 | 0.4457 | 0.6387 | 0.1122 | 0.7183 | 0.1203 | 0.6831 | 0.1592 | 0.2032 | 0.6356 | 0.1429 | 0.7131 | 0.2267 | 0.6705 | 0.1945 | 0.6553 | 0.0773 | 0.6948 | 0.2778 | 0.7151 | 0.2778 | 0.6757 | 0.1355 | 0.6729 | 0.1973 | 0.1559 | 2.0490 | 0.6714 | 0.4459 | 0.7407 | 0.3764 | 0.7062 | 0.7049 | 0.8997 | 0.4423 |
| TRADE | X | 0.8038 | 0.3821 | 0.6272 | 0.1025 | 0.6746 | 0.1102 | 0.6488 | 0.1465 | 0.1890 | 0.6117 | 0.1362 | 0.6797 | 0.2127 | 0.6520 | 0.1803 | 0.6350 | 0.0732 | 0.6705 | 0.2626 | 0.6848 | 0.2728 | 0.6523 | 0.1260 | 0.6489 | 0.1874 | 0.1288 | 2.1062 | 0.5922 | 0.4009 | 0.6572 | 0.3256 | 0.6249 | 0.6229 | 0.8618 | 0.3627 |
| TRADE | X | 0.7261 | 0.2440 | 0.5929 | 0.0942 | 0.6307 | 0.1010 | 0.6649 | 0.1654 | 0.2096 | 0.5725 | 0.1282 | 0.6432 | 0.2389 | 0.6131 | 0.1952 | 0.5975 | 0.0786 | 0.6292 | 0.2843 | 0.6417 | 0.2975 | 0.6014 | 0.1361 | 0.6064 | 0.1992 | 0.1660 | 2.1827 | 0.7051 | 0.5030 | 0.7821 | 0.4299 | 0.7501 | 0.7437 | 0.8039 | 0.2645 |
| EHRmamba | X | 0.8724 | 0.5377 | 0.6428 | 0.1083 | 0.6958 | 0.1169 | 0.6566 | 0.1557 | 0.1981 | 0.6117 | 0.1397 | 0.6820 | 0.2241 | 0.6468 | 0.1892 | 0.6324 | 0.0763 | 0.6766 | 0.2721 | 0.6945 | 0.2851 | 0.6576 | 0.1320 | 0.6550 | 0.1948 | 0.1477 | 2.0724 | 0.6553 | 0.4540 | 0.7234 | 0.3749 | 0.6942 | 0.6895 | 0.8932 | 0.4401 |
| EHRmamba | X | 0.8109 | 0.3917 | 0.6577 | 0.0997 | 0.6577 | 0.1074 | 0.6279 | 0.1444 | 0.1865 | 0.5845 | 0.1330 | 0.6509 | 0.2241 | 0.6192 | 0.1892 | 0.6045 | 0.0763 | 0.6458 | 0.2721 | 0.6582 | 0.2851 | 0.6212 | 0.1320 | 0.6224 | 0.1948 | 0.1477 | 2.1357 | 0.5774 | 0.3862 | 0.6410 | 0.3109 | 0.6110 | 0.6093 | 0.8398 | 0.3532 |
| EHRmamba | X | 0.7194 | 0.2493 | 0.5786 | 0.0912 | 0.6105 | 0.0965 | 0.5891 | 0.1298 | 0.1560 | 0.5569 | 0.1247 | 0.6197 | 0.2043 | 0.5934 | 0.1777 | 0.5806 | 0.0732 | 0.6126 | 0.2601 | 0.6218 | 0.2695 | 0.5827 | 0.1266 | 0.5855 | 0.1879 | 0.1203 | 2.2136 | 0.6912(?) | 0.3862 | 0.6410 | 0.3109 | 0.6110 | 0.6093 | 0.7751 | 0.2608 |
| Ours (No value share) | X | 0.8911 | 0.6022 | 0.7458 | 0.1487 | 0.8153 | 0.1835 | 0.7752 | 0.2361 | 0.2828 | 0.7321 | 0.1923 | 0.8102 | 0.3217 | 0.7718 | 0.2630 | 0.7511 | 0.1003 | 0.8027 | 0.3739 | 0.8272 | 0.3951 | 0.7804 | 0.1809 | 0.7756 | 0.2687 | 0.1974 | 1.9608 | 0.7739 | 0.5744 | 0.8490 | 0.4807 | 0.8126 | 0.8084 | 0.9186 | 0.4978 |
| Ours (No value share) | X | 0.8685 | 0.5253 | 0.6987 | 0.1334 | 0.7857 | 0.1705 | 0.7432 | 0.2181 | 0.2627 | 0.7050 | 0.1780 | 0.7731 | 0.2945 | 0.7396 | 0.2463 | 0.7172 | 0.0954 | 0.7637 | 0.3537 | 0.7884 | 0.3748 | 0.7436 | 0.1679 | 0.7351 | 0.2506 | 0.1803 | 2.0212 | 0.7397 | 0.5386 | 0.8041 | 0.4235 | 0.7695 | 0.7652 | 0.8925 | 0.4328 |
| Ours (No value share) | X | 0.8110 | 0.4084 | 0.6523 | 0.1190 | 0.7208 | 0.1537 | 0.6831 | 0.1994 | 0.2403 | 0.6496 | 0.1627 | 0.7129 | 0.2625 | 0.6797 | 0.2193 | 0.6624 | 0.0893 | 0.7014 | 0.3274 | 0.7337 | 0.3459 | 0.6865 | 0.1543 | 0.6810 | 0.2296 | 0.1628 | 2.0943 | 0.6912 | 0.4935 | 0.7557 | 0.3742 | 0.7210 | 0.7167 | 0.8469 | 0.3592 |
| FM4EHR | O | 0.6221 | 0.1842 | 0.5000 | 0.0731 | 0.5000 | 0.0831 | 0.5000 | 0.1248 | 0.1656 | 0.5000 | 0.1097 | 0.5000 | 0.1816 | 0.5000 | 0.1532 | 0.5000 | 0.0601 | 0.5000 | 0.2308 | 0.5000 | 0.2415 | 0.5000 | 0.0290 | 0.5000 | 0.1607 | 0.1183 | 2.2017 | 0.5000 | 0.3136 | 0.5000 | 0.2102 | 0.5000 | 0.5000 | 0.5000 | 0.1651 |
| FM4EHR | O | 0.5678 | 0.1427 | 0.5000 | 0.0731 | 0.5000 | 0.0831 | 0.5000 | 0.1248 | 0.1656 | 0.5000 | 0.1097 | 0.5000 | 0.1816 | 0.5000 | 0.1532 | 0.5000 | 0.0601 | 0.5000 | 0.2308 | 0.5000 | 0.2415 | 0.5000 | 0.0290 | 0.5000 | 0.1607 | 0.1139 | 2.2291 | 0.5000 | 0.3136 | 0.5000 | 0.2102 | 0.5000 | 0.5000 | 0.5000 | 0.1651 |
| FM4EHR | O | 0.6923 | 0.1118 | 0.5000 | 0.0731 | 0.5000 | 0.0831 | 0.5000 | 0.1248 | 0.1656 | 0.5000 | 0.1097 | 0.5000 | 0.1816 | 0.5000 | 0.1532 | 0.5000 | 0.0601 | 0.5000 | 0.2308 | 0.5000 | 0.2415 | 0.5000 | 0.0290 | 0.5000 | 0.1607 | 0.0912 | 2.2807 | 0.5000 | 0.3136 | 0.5000 | 0.2102 | 0.5000 | 0.5000 | 0.5000 | 0.1651 |
| ETHOS | O | 0.8607 | 0.5290 | 0.6809 | 0.1079 | 0.7441 | 0.1251 | 0.7066 | 0.1746 | 0.1936 | 0.6604 | 0.1376 | 0.7369 | 0.2517 | 0.6960 | 0.1973 | 0.6798 | 0.0839 | 0.7107 | 0.2825 | 0.6601 | 0.2791 | 0.6951 | 0.1360 | 0.6920 | 0.1979 | 0.1708 | 2.0134 | 0.6684 | 0.4578 | 0.7390 | 0.4407 | 0.7101 | 0.7068 | 0.8979 | 0.3605 |
| ETHOS | O | 0.8276 | 0.4481 | 0.6422 | 0.0998 | 0.6682 | 0.1160 | 0.6669 | 0.1629 | 0.1952 | 0.6248 | 0.1311 | 0.6698 | 0.2224 | 0.6332 | 0.1872 | 0.6184 | 0.0731 | 0.6468 | 0.2664 | 0.6658 | 0.2817 | 0.6245 | 0.1293 | 0.6282 | 0.1885 | 0.1529 | 2.1802 | 0.6166 | 0.4106 | 0.6812 | 0.3245 | 0.6439 | 0.6420 | 0.8586 | 0.2749 |
| ETHOS | O | 0.7559 | 0.3306 | 0.6120 | 0.1003 | 0.6732 | 0.1098 | 0.6289 | 0.1531 | 0.1851 | 0.5880 | 0.1249 | 0.6489 | 0.2130 | 0.6132 | 0.1810 | 0.5989 | 0.0716 | 0.6297 | 0.2557 | 0.6413 | 0.2693 | 0.6291 | 0.1302 | 0.6298 | 0.1892 | 0.1254 | 2.1628 | 0.6131 | 0.4070 | 0.6818 | 0.3220 | 0.6426 | 0.6402 | 0.7942 | 0.2717 |
| StraTS | O | 0.7610 | 0.3076 | 0.6146 | 0.0921 | 0.6408 | 0.1015 | 0.6310 | 0.1554 | 0.1769 | 0.5930 | 0.1311 | 0.6694 | 0.2209 | 0.6332 | 0.1860 | 0.6184 | 0.0734 | 0.6468 | 0.2678 | 0.6658 | 0.2817 | 0.6055 | 0.1254 | 0.6046 | 0.1821 | 0.1280 | 2.2125 | 0.5820 | 0.3752 | 0.6410 | 0.2940 | 0.6074 | 0.6053 | 0.7601 | 0.2325 |
| StraTS | O | 0.7088 | 0.2459 | 0.5859 | 0.0874 | 0.6069 | 0.0963 | 0.6031 | 0.1479 | 0.1769 | 0.5724 | 0.1190 | 0.6239 | 0.1979 | 0.6132 | 0.1680 | 0.5735 | 0.0675 | 0.6297 | 0.2397 | 0.6413 | 0.2539 | 0.5706 | 0.1197 | 0.5721 | 0.1732 | 0.1097 | 2.2547 | 0.5523 | 0.3437 | 0.6072 | 0.2693 | 0.5766 | 0.5751 | 0.7238 | 0.1980 |
| StraTS | O | 0.6920 | 0.2197 | 0.5632 | 0.0860 | 0.5857 | 0.0890 | 0.5747 | 0.1404 | 0.1660 | 0.5481 | 0.1140 | 0.5900 | 0.1804 | 0.5900 | 0.1590 | 0.5500 | 0.0640 | 0.5992 | 0.2250 | 0.6000 | 0.2400 | 0.5600 | 0.1140 | 0.5600 | 0.1640 | 0.0924 | 2.2547 | 0.5500 | 0.3200 | 0.6000 | 0.2500 | 0.5600 | 0.5600 | 0.7100 | 0.1800 |
| Ours (Value share) | O | 0.8789 | 0.5617 | 0.7057 | 0.1411 | 0.7925 | 0.1759 | 0.7567 | 0.2389 | 0.2752 | 0.7154 | 0.1870 | 0.7856 | 0.3099 | 0.7504 | 0.2537 | 0.7338 | 0.0979 | 0.7608 | 0.3642 | 0.8049 | 0.3873 | 0.7551 | 0.1768 | 0.7518 | 0.2636 | 0.1890 | 1.9917 | 0.7556 | 0.5560 | 0.8239 | 0.4589 | 0.7903 | 0.7859 | 0.9115 | 0.4762 |
| Ours (Value share) | O | 0.8540 | 0.4998 | 0.6642 | 0.1269 | 0.7510 | 0.1638 | 0.7186 | 0.2095 | 0.2546 | 0.6802 | 0.1720 | 0.7507 | 0.2859 | 0.7135 | 0.2354 | 0.6956 | 0.0919 | 0.7427 | 0.3426 | 0.7680 | 0.3647 | 0.7217 | 0.1626 | 0.7142 | 0.2444 | 0.1708 | 2.0498 | 0.7185 | 0.5165 | 0.7813 | 0.4072 | 0.7486 | 0.7441 | 0.8756 | 0.4058 |
| Ours (Value share) | O | 0.7948 | 0.3864 | 0.6230 | 0.1154 | 0.6965 | 0.1490 | 0.6588 | 0.1901 | 0.2300 | 0.6244 | 0.1564 | 0.6912 | 0.2619 | 0.6607 | 0.2154 | 0.6456 | 0.0852 | 0.6780 | 0.3128 | 0.7089 | 0.3332 | 0.6597 | 0.1472 | 0.6588 | 0.2278 | 0.1529 | 2.1220 | 0.6631 | 0.4680 | 0.7278 | 0.3568 | 0.6943 | 0.6899 | 0.8230 | 0.3229 |

## G   LLM USAGE CLARIFICATION

In addition to the uses of LLMs described in the main text, we employed them for summarizing content, translation, grammar correction, and sentence refinement during the writing of the manuscript. In the early stages of the study, we used LLMs to search for related work, and the retrieved papers were then read and verified by the researchers.

**Ours (share)**

Birth : 1845. 05. 11
Sex: Female
Ethnicity: WHITE
Age: 300

2145-05-11 17:12:55
- ICU transfer
2145-05-11 17:30:00
- DBP : 62
- SBP : 103
- HR : 123
- MBP : 83
- RR : 4
2145-05-11 18:00:00
- DBP : 51
- HR : 110
- MBP : 73
- RR : 17
- SBP : 111
—-# Gen Start #—-
- O2 saturation : 90
- Weight : 54.2
- Temperature : 36.7
- GCS : 15
- GCS-M : 6
- GCS-E : 4
- GCS-V : 5
- O2 saturation : 92
- HR : 117
- SBP : 108
- Glucose : 261
- MBP : 76
- DBP : 50
- Anion gap : 15.0
- Bicarbonate : 24.0
- Calcium Total : 8.6
- Chloride : 105.0
- Creatinine : 0.9
- Magnesium : 1.8
- Phosphate : 2.7
- Potassium : 4.0
- Sodium : 142.0
- Blood urea nitrogen
: 12.0
- Hematocrit : 33.0
- Hemoglobin chem-
istry : 11.3
- Prothrombin time
INR : 1.10
- Mean corpuscular

hemoglobin : 31.0
- Mean corpuscular
hemoglobin concen-
tration : 34.4
- Mean corpuscular
volume : 90.0
- Platelets : 200.0
- Prothrombin time :
13.3
- Partial thromboplas-
tin time : 32.4
- Red Cell Distribu-
tion Width : 13.3
- Red blood cell count
: 3.74
- White blood cell
count : 12.20
- RR : 18
2145-05-11 19:00:00
- DBP : 56
- HR : 119
- MBP : 73
- O2 saturation : 92
- RR : 19
- SBP : 111
2145-05-11 20:00:00
- DBP : 50
- MBP : 76
- HR : 110
- RR : 19
- SBP : 111
- O2 saturation : 90
2145-05-11 21:00:00
- DBP : 51
- HR : 113
- MBP : 71
- O2 saturation : 93
- RR : 19
- SBP : 115
2145-05-11 22:00:00
- DBP : 50
- HR : 114
- MBP : 74
- O2 saturation : 89
- RR : 19
- SBP : 116
2145-05-11 23:00:00
- DBP : 51
- GCS-E : 4
- GCS-M : 6

- GCS : 15
- GCS-V : 5
- HR : 117
- MBP : 73
- O2 saturation : 92
- RR : 19
- SBP : 115
- Temperature : 36.9
2145-05-12 00:00:00
- SBP : 116
- RR : 19
- O2 saturation : 91
- HR : 118
- DBP : 53
- MBP : 74
2145-05-12 01:00:00
- DBP : 50
- HR : 116
- MBP : 76
- O2 saturation : 93
- RR : 18
- SBP : 113
2145-05-12 02:00:00
- HR : 111
- DBP : 49
- MBP : 74
- RR : 20
- SBP : 116
- O2 saturation : 92
2145-05-12 03:00:00
- DBP : 52
- GCS-E : 4
- GCS-M : 6
- GCS : 15
- GCS-V : 5
- HR : 119
- MBP : 72
- O2 saturation : 91
- RR : 20
- SBP : 109
- Temperature : 36.9
2145-05-12 04:00:00
- DBP : 50
- HR : 117
- MBP : 73
- RR : 18
- SBP : 113
- O2 saturation : 85
2145-05-12 05:00:00
- DBP : 49

- HR : 116
- MBP : 72
- O2 saturation : 79
- RR : 19
- SBP : 116
2145-05-12 06:00:00
- DBP : 47
- HR : 114
- MBP : 70
- O2 saturation : 92
- RR : 20
- SBP : 107
2145-05-12 07:00:00
- HR : 111
- DBP : 49
- GCS-E : 4
- GCS-M : 6
- GCS : 14
- GCS-V : 5
- MBP : 73
- O2 saturation : 93
- RR : 19
- SBP : 114
- Temperature : 36.6
2145-05-12 08:00:00
- DBP : 52
- HR : 112
- MBP : 71
- O2 saturation : 89
- RR : 19
- SBP : 114
2145-05-12 09:00:00
- DBP : 52
- HR : 116
- MBP : 76
- O2 saturation : 88
- RR : 20
- SBP : 110
2145-05-12 10:00:00
- DBP : 53
- HR : 113
- MBP : 75
- O2 saturation : 92
- RR : 20
- SBP : 111
2145-05-12 11:00:00
- DBP : 52
- GCS-E : 4
- GCS-M : 6
...

Figure 8: Temporal EHR history generated by our model (used value share version for fair compar-
ison). The data before this "—# Gen Start #—" marker is given, and the data after it is generated.

**ETHOS**

Birth : 1845. 05. 11
Sex: Female
Ethnicity: WHITE
Age: 300

2145-05-11 17:12:55
- ICU transfer
2145-05-11 17:30:00
- DBP : 62
- SBP : 103
- HR : 123
- MBP : 83
- RR : 4
2145-05-11 18:00:00
- DBP : 51
- HR : 110
- MBP : 73
- RR : 17
- SBP : 111
—-# Gen Start #—-
- O2 saturation : 97
2145-05-11 18:30:00
- DBP : 51
- HR : 124
- MBP : 67
- O2 saturation : 97
- RR : 32
- SBP : 89
2145-05-11 19:10:00
- SBP : 80
- RR : 35
- O2 saturation : 96
- HR : 119
- DBP : 42
- MBP : 59
2145-05-11 19:45:00
- DBP : 44
- HR : 129
- MBP : 60
- O2 saturation : 90
- RR : 30
- SBP : 94
2145-05-11 20:15:00
- RR : 32
- SBP : 78
- O2 saturation : 93
- HR : 140

- DBP : 42
- MBP : 60
2145-05-11 20:45:00
- RR : 30
- SBP : 84
- O2 saturation : 93
- HR : 115
- DBP : 44
- MBP : 49
2145-05-11 21:25:00
- HR : 122
- MBP : 58
- O2 saturation : 93
- RR : 29
- SBP : 88
- DBP : 40
2145-05-11 22:05:00
- SBP : 81
- RR : 29
- O2 saturation : 87
- HR : 134
- DBP : 43
- MBP : 59
2145-05-11 23:00:00
- SBP : 88
- RR : 36
- O2 saturation : 91
- HR : 115
- DBP : 22
- MBP : 58
2145-05-11 23:30:00
- SBP : 84
- RR : 32
- O2 saturation : 92
- HR : 137
- DBP : 38
- MBP : 49
2145-05-11 23:45:00
- DBP : 39
- HR : 115
- MBP : 52
- O2 saturation : 91
- RR : 34
- SBP : 86
2145-05-12 00:15:00
- SBP : 88
- RR : 36

- O2 saturation : 85
- HR : 119
- DBP : 43
- MBP : 60
2145-05-12 00:30:00
- RR : 32
- HR : 114
- O2 saturation : 93
- SBP : 68
- MBP : 43
- DBP : 43
2145-05-12 01:00:00
- DBP : 42
- HR : 117
- MBP : 53
- O2 saturation : 92
- RR : 31
- SBP : 92
2145-05-12 01:50:00
- HR : 112
- O2 saturation : 90
- RR : 33
2145-05-12 02:40:00
- DBP : 39
- MBP : 48
- SBP : 91
2145-05-12 03:15:00
- DBP : 43
- HR : 120
- MBP : 43
- O2 saturation : 91
- RR : 38
- SBP : 94
2145-05-12 04:05:00
- RR : 29
- O2 saturation : 91
- HR : 116
- MBP : 51
- DBP : 38
- SBP : 88
2145-05-12 04:20:00
- RR : 30
- O2 saturation : 87
- HR : 123
- DBP : 42
- MBP : 54
- SBP : 89

2145-05-12 05:05:00
- HR : 115
- O2 saturation : 93
- RR : 29
2145-05-12 05:25:00
- DBP : 31
- MBP : 59
- SBP : 94
2145-05-12 06:25:00
- RR : 33
- O2 saturation : 92
- HR : 120
2145-05-12 06:45:00
- DBP : 27
- MBP : 55
- SBP : 83
2145-05-12 07:35:00
- HR : 119
- O2 saturation : 91
- RR : 29
2145-05-12 08:05:00
- DBP : 42
- MBP : 42
- SBP : 86
2145-05-12 09:05:00
- HR : 116
- MBP : 50
- O2 saturation : 92
- RR : 37
- SBP : 81
- DBP : 43
2145-05-12 10:00:00
- HR : 132
- O2 saturation : 92
- RR : 33
2145-05-12 10:15:00
- DBP : 35
- MBP : 55
- SBP : 94
2145-05-12 11:10:00
- DBP : 43
- HR : 136
- MBP : 34
...

Figure 9: Temporal EHR history generated by ETHOS. The data before this "—# Gen Start #—" marker is given, and the data after it is generated.

You are a physician with extensive ICU experience and an AI researcher familiar with text generation models, such as LLMs.

In this survey, you will compare the quality of EHR texts generated by two different models from the same initial patient history.

The quality of an EHR depends on whether the right clinical events occur at the right times. Please consider both the timing of events and the appropriateness of the events themselves.

First, you will see a few sample ICU EHR texts. Then, for each pair of generated candidates (A and B), you will be asked to decide which one appears more realistic.

<Sample EHR texts>
1. ## Sample 1 ##

2. ## Sample 2 ##

3. ## Sample 3 ##

<end of EHR samples>

<Evaluation candidate A>
## **ETHOS** generated Sample (Random order; Ours can be candidate A) ##

<Evaluation candidate B>
## **Ours** generated Sample (Random order; ETHOS can be candidate B) ##

<Compare two candidates>

Figure 10: LLM input prompt for generated EHR evaluation. We compared the generative performance of our model and ETHOS on LLMs with this prompt.

