# OpenReview forum: "Time Conditioned Foreseeing: Temporal Generative Pretraining for EHR foundation models"
_ICLR.cc/2026/Conference — Submitted to ICLR 2026_

### Official Review · Reviewer_FPVU · 2025-10-27

**Soundness:** 1
**Presentation:** 3
**Contribution:** 1
**Rating:** 2
**Confidence:** 5

**Summary:**

This paper improves EHR foundation models by better featurizing time and numeric values and incorporating time into the pertraining objective.

**Strengths:**

- MIMIC-III is a good starting point for evaluation as it is reproducible (but as noted in weaknesses, I don't think it's sufficient by itself).
- The set of evaluation tasks is good.
- Calendar time features are generally a good idea (except for using the year as a feature). In theory, these features are unnecessary (since they can be learned from ages with neural network). Still, the transformation from ages to calendar time is non-trivial, and giving the model a shortcut might be an excellent idea.

**Weaknesses:**

Experimental Setup / Soundness Weaknesses:
- This model appears to use the year of the visit as a feature, but it is not evaluated correctly. In order to evaluate the effectiveness of absolute time features, you must perform a time split (with held-out test years) to assess whether your trained use of absolute time generalizes to the future. Otherwise, you face severe overfitting issues.
- I would argue that MIMIC-III is too small a dataset for modern foundation model work. Especially when datasets like MIMIC-IV are available.
- The lack of a gradient tree baseline (xgboost or lightgbm) is very concerning. Those baselines are essential for ensuring more sophisticated models are set up correctly, as they perform very well and require minimal tuning.
- Qualitative evaluation of temporal generative modeling is very flawed. Following prior work (see https://www.nature.com/articles/s41746-024-01235-0), the proper procedure is to instead evaluate temporal generation by comparing the output timelines to your predictive models. Out of the patients who died, ETHOS's generated timelines predicted death x% of the time while our model's timelines predicted death y% of the time. If you want to measure the timeliness of the timelines, measure how well the models output death tokens within a certain time window.
- The lack of public code limits my ability to more properly review this paper.
- The summarized test set is insufficient for evaluating ablations in table 4. Test loss is probablemtaic because it is very heavily driven by outliers. Measuring AUROC and AUPRC would be much more reliable.
- How were baselines tuned for hyperparameters? This is especially important with the lack of a gradient tree based baseline (which still requires tuning, but is much less sensitive).

Related Work / Contribution Weaknesses
- The Table 1 (literature review) table has many flaws. EHRMamba and EHRSHOT all use numeric values with quantile binning. A lot of the models also use relative intervals. I know Foresight, EHRMamba and EHRSHOT at minimum use them.
- The use of the word "uniform" numeric binning is not accurate for describing the prior work. The prior work being compared to uses quantile binning, which is very non-uniform.
- I don't understand the non-concurrency column in table 1 and in your contributions. Isn't non-concurrency implicit with any relative time featurization? How can a model have relative time input but be missing non-concurrency?
- EventStreamGPT should really be mentioned in related work, as it focuses on time conditioned pretraining with EHR data.
- The marks for temporal generation appears to be wrong for many of the entries in the literature review. For example, Foresight supports temporal generation.
- The novelty of the time conditioned forcasting objective is unclear. Especially compared to related work like EventStreamGPT. I would argue ETHOS also supports it to a similar extent to your model.
- The time-conditioned pretraining for this model appears to require rounding the time to various calendar time bins? And then cross entropy loss is taken? How is this different from ETHOS's time tokenization? AKA it's not a real continuous time prediction model like EventStreamGPT?

**Questions:**

See weaknesses section.

---

> ### Author Response · Authors · 2025-12-02
> **Official Comment by Authors (1/3)**
>
> We sincerely thank the reviewer for taking the time to carefully read our manuscript and for providing valuable suggestions that meaningfully improve the paper.
>
> We also appreciate your positive feedback acknowledging the strength of our evaluation setup and the thoughtful use of calendar-time features. In particular, we are grateful that you recognized our design choice as meaningful — noting that although neural networks could theoretically infer such information, providing an explicit shortcut may be a practical and effective idea.
>
> ---
>
> First, your detailed comments were highly helpful to us. We acknowledge that there are aspects of this work that can be improved, and we aim to follow many of your recommendations, including:
>
> - Adding experiments based on MIMIC-IV
> - Adding gradient tree baselines
> - Providing quantitative temporal generation evaluation
> - Including EventStreamGPT in the related work section
> - Running ablation studies based on AUROC instead of test loss
>
> We also provide the following clarifications.
>
> As stated in Appendix D.2, we conducted hyperparameter search for the learning rate, and all remaining training setups were kept identical across models.
>
> We acknowledge that using the word *“uniform”* numeric binning was inappropriate, and we appreciate your comment. Our intention was to highlight that quantile binning makes the **sample count uniform across bins**, effectively transforming the data into a **uniform distribution**, even though bin widths vary. We have revised this phrasing to: *“quantile binning, which equalizes the number of samples per bin,”* to more accurately reflect the prior work.
>
> We also discovered and corrected an error in Table 1 and the Related Work section, where we had mistakenly stated that EHRSHOT does not address relative temporal intervals, while in fact it does.
> However, we clarify that **Foresight and EHRMamba do not address relative temporal intervals.**
>
> To reproduce EHRSHOT, we used the femr package with the model `StanfordShahLab/clmbr-t-base-random` (i.e., the implementation at `femr/models/transformer.py`; femr package: https://github.com/som-shahlab/femr), and therefore our experimental results remain valid.

---

> ### Author Response · Authors · 2025-12-02
> **Official Comment by Authors (2/3)**
>
> However, for some of your comments, it seems that you may have a misunderstanding of the related work and our paper. Therefore, we provide the following clarifications.
>
> ---
>
> > First, regarding your comment, “EHRMamba and EHRSHOT all use numeric values with quantile binning,” we have already noted in Table 1 that, as you stated, both models use numeric values with quantile binning. We believe this was a misreading.
>
> ### 1. What it means for a model to address relative time intervals and why it matters
> We adopted this concept from **well-established positional embeddings in Transformers**. Positional embeddings inject "position information" for each token. In EHR models where each token represents an event, this corresponds to the event's position on the timeline.
> Early Transformers (Vaswani et al., 2017) with sin/cos positional embeddings, or GPT-2 with learned positional embeddings, are examples of **Absolute Positional Embeddings (APE)**. Rotary positional embeddings are a representative form of **Relative Positional Embeddings (RPE)** and are now commonly used in many LLMs.
> Relative positional embeddings mean that the attention module referencing the i-th token X to the j-th token Y is a function of f(X, Y, (i-j)).
> RPEs are well-known to offer clear advantages over APEs in **generalization**, particularly in **length extrapolation** (handling sequences longer than those seen in training) and **translation invariance**.
>
> ### 2. Why our model addresses relative temporal intervals
> As mentioned above, RPE is a function of relative position i-j. When we convert token positions into a temporal context, this becomes a function of the time gap (ΔT) between attended events. Our methodology applies calendrical periodic RoPE to the keys and queries in the attention module, thereby fulfilling this criterion and allowing the model to address relative temporal intervals.
>
> ### 3. Why Foresight and EHRMamba do not address relative intervals
> First, **EHRMamba** (https://arxiv.org/pdf/2405.14567, Section 2.2) clearly uses absolute positional embeddings. Appendix C.2 shows that its time embedding is a function of age using Time2Vec. Time2Vec uses a formulation very similar to the Transformer's original absolute positional embeddings (sinusoidal embeddings). While it transforms time into a periodic representation, it remains an absolute PE, making it vulnerable in terms of generalization.
>
> Second, **Foresight** (https://arxiv.org/pdf/2212.08072) uses more basic temporal information. On page 6, it describes itself as a GPT-2-based Generative Pretrained Transformer using learned positional embeddings. This model inserts an age token upon reaching the corresponding age and adds a \<SEP\> token at the start of each day. This treats all events within a day as occurring at the same time, and thus cannot address relative intervals; the temporal gaps between events are neither provided to the model during training nor generated by it.
>
> ### 4. Regarding the suggested experiment on year-based OOD testing
>
> Your suggestion — *“to evaluate absolute time features, you must perform a split with held-out test years”* — is theoretically unnecessary.
>
> We apologize if our writing caused confusion. Our temporal positional embedding is invariant to yearly shifts. Therefore, using this model on future data does not cause generalization issues.
> By using an anchor token at January 1st, 00:00:00 (line 1167), the model addresses absolute timestamps **within a year**, such as seasonal or day/night variations. Even if all data were shifted by +10 years, the model’s computations would remain identical. This is analogous to a RoPE-applied model producing the same outputs regardless of the starting token position.

---

> ### Author Response · Authors · 2025-12-02
> **Official Comment by Authors (3/3)**
>
> ### 5. On non-concurrency
> As described in our caption, this refers to whether a model can distinguish concurrent events. Transformer encoders are permutation-invariant, so without positional embeddings, they cannot recognize order. (For Transformer decoders, causal masks act as NoPE, but positional embeddings to differentiate tokens remain important.)
> When positional embeddings are based on event timestamps, events that occur at the same timestamp become permutation-invariant. Non-concurrency refers to whether there exists a mechanism to distinguish such events occurring simultaneously (as mentioned in line 100 of the main text).
> For example, for a patient with a fever, a blood culture test and antibiotic administration may be ordered concurrently, but the blood culture must occur first. Moreover, although beyond the scope of this paper, when extending the EHR modality to text, non-concurrency becomes important for encoding texts recorded at the specific timestamp into the model.
>
> ### 6. Why ETHOS is limited, and how our model differs
> **Time-conditioned pretraining in our model does not require rounding time to various calendar bins.** This is one of the novelties of our work. For example, if **Event 1** occurs at *2025-01-01 00:00:00* and **Event 2** occurs at *2025-04-15 08:12:00*, our model uses dual-calendar RoPE to accurately encode **Event 1** and learn the exact interval to the next task: 0 years, 3 months, 14 days, 8 hours, 12 minutes. In contrast, ETHOS does not use the event timestamp as input; it simply generates a "3–6 month" token.
>
> **Event B occurs 2–6 hours after Event A, and Event C occurs 5–15 minutes after B**: this is how **ETHOS** performs temporal generation. **Our model**, on the other hand, generates the **precise occurrence time of each event** down to minutes (seconds can also be generated, but we match the dataset's granularity).
>
> **ETHOS** has several limitations:
> a. **Time information is discrete and quantized, with larger errors for longer intervals**. The smallest interval is 5–15 minutes, and the largest is 3–6 months. This induces significant errors and reduces clinical applicability. For example, if a token indicates 6–12 hours, it is unclear whether the event occurs in 6 or 12 hours, which is critical for medication timing.
> b. **Errors accumulate over multiple intervals**. If the sequence A/2–6 hr/B/5–15 min/C is generated, the total interval between A and C could be 2 hr 5 min to 6 hr 15 min, making the generated information unreliable.
> c. **To attend to the time interval between two events, the model must attend to all intermediate time tokens**. LLMs are known to train attention sparsely across some tokens, and this constraint can degrade performance on long EHR sequences.
>
> In contrast, our model uses a continuous time representation, generating precise timestamps for each event. Because time is included as a positional embedding, attending to only the tokens I and J is sufficient to determine the interval between events I and J.
>
> ### 7. Foresight does not support temporal generation
> Temporal generation refers to generating full events, including Time-Feature-Value, from the model output. However, Foresight approximates event times at a daily resolution using the \<SEP\> token, which lacks clinical meaning. The \<SEP\> token generation in Foresight can be considered a lower-resolution version of ETHOS using 13 time tokens.
>
> ### 8. Comparison with EventStreamGPT (ESGPT)
> First, **ESGPT uses Absolute Positional Embeddings for time** (Supplementary Material Section 6.2; https://openreview.net/forum?id=hiO0735tmc&referrer=%5Bthe%20profile%20of%20Bret%20Nestor%5D(%2Fprofile%3Fid%3D~Bret_Nestor1)), referred to as "Sinusoidal Absolute Temporal Embeddings" in the paper. Even if Events A, B, and C occur at equal intervals, the attention functions from B→A and C→B differ, which limits generalization.
> Second, temporal generation in **ESGPT uses pre-defined parameterized probability density functions** to compute delta T from the last event, severely limiting temporal modeling capability. There is no justification for using pre-defined Exponential or Mixture-of-LogNormals distributions for all clinical events. For example, measuring vital signs every 8 hours and the probability of catching a cold should not share the same distribution, even with different parameters—a primitive approach. (In contrast, the MOTOR paper reasonably approximates disease incidence in limited outpatient scenarios using a Poisson distribution.)
> **c.f.** Our model performs direct DNN-based modeling without assuming any distribution.
> Third, **ESGPT also relies on the next event**, and therefore cannot perform long-term prediction beyond this dependency.
> These findings indicates that ESGPT is not relevant to the novelty of our work.
>
> ---
>
> Once again, we sincerely appreciate your contributions and valuable comments.

---

### Official Review · Reviewer_7R5L · 2025-10-31

**Soundness:** 3
**Presentation:** 4
**Contribution:** 3
**Rating:** 6
**Confidence:** 4

**Summary:**

The paper proposes a new temporal generative pretraining framework for Electronic Health Record foundation models, called Time-Conditioned Foreseeing. The model introduces three main innovations: a Pathology-Focused Binning method for emphasizing clinically significant numeric ranges, a Dual-Calendar Rotary Positional Embedding for encoding both absolute and relative time, and a Time-Conditioned Foreseeing objective for forecasting future clinical events at multiple temporal horizons. The model is trained and evaluated on the MIMIC-III dataset, showing improved performance across seven downstream clinical prediction tasks, with up to 48% higher AUPRC than prior EHR foundation models. Experiments further show that the model generates realistic and temporally consistent synthetic EHRs that capture physiological time patterns such as circadian variations. The authors conclude that their approach establishes the first genuine temporal generative EHR model and identify a lack of established metrics for evaluating temporal EHR generation as a key limitation.

**Strengths:**

Below are the strengths of the paper in my opinion:

1. The paper identifies a clear problem in how prior EHR foundation models rely on NLP-style pretraining and motivates its approach by emphasizing the temporal and numerical nature of EHR data, making the contribution well justified.
2. It introduces three specific techniques (Pathology-Focused Binning, Dual-Calendar RoPE, and Time-Conditioned Foreseeing) with mathematical definitions and clear intuition, showing conceptual clarity and novelty.
3. The model consistently outperforms all baselines across seven downstream tasks on MIMIC-III and demonstrates up to 48% higher AUPRC, supporting the strength of its results.
4. An ablation study verifies that removing or replacing any of the proposed components degrades performance, which strengthens the paper’s claims.
5. The use of the public MIMIC-III dataset helps with transparency and reproducibility although no code is provided as supplementary material yet.
6. The visualizations of the paper are great. Visualizations such as the circadian rhythm plot provide qualitative evidence that the model generates realistic and temporally consistent EHR sequences.

**Weaknesses:**

Below are the weaknesses of the paper in my opinion:

1. The paper lacks quantitative metrics for evaluating generative realism and acknowledges that no established evaluation exists for temporal EHR generation.
2. All experiments are limited to the MIMIC-III dataset, leaving generalization to other settings untested. Additionally, I do not see why the authors have not used much richer MIMIC-IV dataset. Extending the methods to MIMIC-IV should not be too hard and I think is a clear missing point in the current state of the work.
3. The Time-Conditioned Foreseeing mechanism is densely written and could be difficult to follow, despite including equations and diagrams. Please add more descriptive details to the current appendix section for this part.
4. The paper omits any discussion of computational cost or efficiency implications of its multi-component architecture.
5. There is no analysis of fairness, subgroup behavior, or bias, even though the domain involves clinical data.
6. The limitations of the paper is not discussed well. Failure cases and uncertainty sources are not discussed, leaving the reader without insight into model limitations.

**Questions:**

Most my questions are embedded in the weaknesses section. My main question to improve is use of other diverse EHRs and seeing how the modeling pipeline transfers to datasets other than MIMIC-III. In addition, some extra clarifying details would be helpful for the following minor questions:

1. How does the proposed TCF objective differ in practice from standard next-token prediction when generating long-term event sequences?
2. Can the model handle irregular or missing time intervals in EHR sequences, and if so, how is this addressed in preprocessing or training?
3. What specific criteria were used to select the seven downstream tasks, and how do they relate to the generative capabilities claimed by the paper?
4. Does the Dual-Calendar RoPE generalize to time zones or varying calendar systems, or is it tailored specifically to the MIMIC-III dataset’s timestamp format? This is especially important because across datasets the deanonymization on time could vary and this can contaminate the pipeline.
5. How sensitive is the Pathology-Focused Binning approach to the bandwidth parameter in the Gaussian kernel density estimation, and was any hyperparameter tuning performed?
6. How do the authors ensure that the qualitative evaluations meaningfully reflect real-world clinical plausibility beyond what is currently done? I feel like the interpretability aspect here can be improved.
7. Could the proposed model inadvertently memorize training patients or sequences during generative pretraining, and were any checks for memorization performed?

**Details Of Ethics Concerns:**

n/a.

---

> ### Author Response · Authors · 2025-12-02
>
> We sincerely thank the reviewer for dedicating valuable time to carefully read our manuscript and provide feedback that significantly strengthened our work.
>
> We are also grateful that you recognized the strengths of our paper — specifically that it identifies core limitations in existing EHR foundation models and proposes well-motivated methods tailored to the temporal and numerical characteristics of EHR data, supported by strong empirical performance and comprehensive analysis.
>
> ---
>
> We fully appreciate your comments and intend to address the highlighted weaknesses with the following revisions:
>
> 1. We agree that evaluating generative realism is important, and we will consider incorporating additional metrics (e.g., mortality prediction during generation).
> 2. We will extend the current MIMIC-III experiments to MIMIC-IV and reproduce the full evaluation pipeline.
> 3. We will improve the clarity of writing around the Time-Conditioned Foreseeing mechanism and include additional technical details in the appendix.
> 4. We will compare the computational cost of our model with relevant baselines.
>    *Please note that most computation is shared in the decoder, and multiple time-conditioned forecasting occurs only in the final head.*
> 5. We will conduct additional subgroup analyses (e.g., race- and sex-specific performance) to further assess fairness and robustness.
> 6. We will provide a more detailed discussion of failure cases in EHR generation and strengthen the limitations section.
>
> ---
>
> ### **Clarifications to Your Questions**
>
> 1. Simple next-token prediction cannot model temporal structures necessary for EHR generation. In contrast, our Time-Conditioned Foreseeing mechanism jointly models:
>    - next timestamp prediction: $P(T_{\text{next}} \mid E_{\text{past}})$,
>    - next feature–value prediction: $P(F_{\text{next}}, V_{\text{next}} \mid T_{\text{next}}, E_{\text{past}})$,  enabling coherent sequential event generation.
> Moreover, our model supports arbitrary future forecasting of given timestamp via  $P(F_{\text{future}}, V_{\text{future}} \mid T_{\text{future}}, E_{\text{past}})$.
>
> 2. Dual-Calendar RoPE embeds explicit timestamps, allowing the model to handle irregular intervals and simultaneous events. This requires timestamp information in the dataset; events lacking timestamps were excluded during preprocessing.
>
> 3. We added three additional downstream tasks on top of the all four standard benchmark tasks (*Multitask learning and benchmarking with clinical time series data, 2019*) to ensure evaluation across a broader and clinically diverse spectrum (there was no cherry-picking). Through this wider range of tasks, we demonstrated that generative pretraining enables the model to acquire strong representational capacity. While strong representation does not inherently guarantee strong generative ability, our new generative pretraining objective provides evidence that it helps the model learn key structural and temporal characteristics of EHR data. This observation aligns with findings in language modeling research, where partial correlations between generative capability and downstream task performance have been reported, as shown in prior work (*Improving Language Understanding by Generative Pre-training*, OpenAI and  *Connecting Pre-trained Language Model and Downstream Task via Properties of Representation*, NeurIPS 2023).
>
>
> 4. Since Dual-Calendar RoPE is fundamentally a relative positional encoding mechanism, it is compatible with timestamped EHR data expressed in formats such as `YYYY-MM-DD HH:MM:SS` and remains robust under timezone shifts. Because it encodes events based on their relative temporal intervals, it functions correctly even with datasets such as MIMIC where timestamps are anonymized by year-shifting. However, if anonymization further disrupts circadian or seasonal structure (e.g., shifting hourly or monthly), the benefits of calendrical alignment — such as learning increased cardiovascular risk in winter or elevated cortisol levels in the morning — would be reduced.
>
> 5. We agree that evaluating the parameter sensitivity of the Gaussian kernel density estimation is meaningful and will include this experiment in our revisions. As noted in Appendix D.2, we performed hyperparameter search only on the learning rate, while all remaining configurations were kept consistent across models.
>
> 6. Thank you for highlighting this point. We will include additional quantitative evaluation of generative quality.
>
> 7. While we did not evaluate memorization in the current version, we acknowledge its importance, particularly in realistic deployment settings. Memorization tends to increase with model scale (*Quantifying Memorization Across Neural Language Models, 2023*). Given that our models are relatively small (~20M parameters), we expect the risk to be limited, but we plan to include an analysis in future work.
>
> ---
>
> Once again, we sincerely appreciate your thoughtful and constructive feedback.

---

### Official Review · Reviewer_V1Wk · 2025-11-01

**Soundness:** 3
**Presentation:** 3
**Contribution:** 3
**Rating:** 6
**Confidence:** 4

**Summary:**

The paper focuses on EHR generation, addressing the limitation of over-relying on natural language processing to model numeric intensive and time sensitive EHRs. This paper proposes 1) Pathology-Focused Binning strategy to emphasize clinically significant numerical values with density-based tokenizing, 2) Dual-Calendar Rotary Positional Embedding to consider calendrical context and relative time intervals at the same time, 3) Time-Conditioned Foreseeing learning objective to inform forecasting with explicit temporal horizons. Extensive experiments demonstrate the effectiveness of the proposed method, producing substantially highest results over seven downstream tasks. Human evaluation and LLM judgements both suggest high generation fidelity. Case study shows that the proposed method can correctly separate calendrical relevant and irrelevant values, which further validates the Dual-Calendar RoPE design.

**Strengths:**

1. The paper is well-motivated and is well-positioned within literature. The overall structure is clear, and the writing is easy to follow. Discussion into related work and contributions are thorough.
2. The Pathology-Focused Binning strategy, Dual-Calendar Rotary Positional Embedding and Time-Conditioned Foreseeing learning objective are innovative, each addressing a critical and specific problem of existing EHR foundation model.
3. The effectiveness of the proposed model is evaluated through multiple angles, including test loss, downstream metrics, human evaluation, LLM evaluation and case study. Results demonstrate that the proposed method consistently outperforms baselines.
4. The implementation descriptions contain sufficient details for understanding, and source codes are claimed to be released, so reproducibility is satisfying.

**Weaknesses:**

1. The calendrical period embedding seems to be sparse with exponentially expanding discrete window, which may result in inefficiency.
2. The two-stage manner and multi-scale forecasting in time-conditioned foreseeing consume extra computes.
3. The ablation study is inadequate, presenting only test loss results instead of downstream task-specific evaluation metrics which is of higher clinical significance.
4. Minor formatting issue: the equations are not numbered.

**Questions:**

1. How is value-sharing implemented? Why do methods with shared value perform worse than methods without value-sharing?
2. Is the density-based binning strategy robust to OOD values? Will there be any potential distribution shift problem in the experimental or real clinical scenario, and how does this method handle it?
3. Is the performance sensitive to model size? I have noted that some baseline models (e.g., MOTOR, ETHOS) are originally designed to contain more trainable parameters than in this paper, and some results in the original papers show higher performance (e.g. ETHOS obtains an AUC of 0.921 in IHM task.)

---

> ### Author Response · Authors · 2025-12-02
>
> Reviewer 3
>
> We sincerely thank the reviewer for taking the time to thoroughly read our manuscript and for providing valuable suggestions that can significantly improve the quality of our work.
>
> We are also grateful for your positive evaluation recognizing the core strengths of our paper: it features innovative contributions such as Pathology-Focused Binning, Dual-Calendar Rotary Positional Embedding, and Time-Conditioned Foreseeing, and is thoroughly evaluated through test loss, downstream metrics, human and LLM-based assessments, and case studies, consistently outperforming baselines. We greatly appreciate your recognition of these contributions.
>
> Following your feedback and that of the other reviewers, we plan to **strengthen the Related Work section, clarify our methodology, and enrich the paper by incorporating additional training datasets (MIMIC-IV)**.
>
> **[W1]** Unlike natural language, where tokens are modeled with uniform intervals, EHR events can range from minutes apart to years apart. Using fixed-time-unit (e.g., seconds) periodic embeddings in this setting leads to extremely sparse representations. In contrast, our calendrical period embedding compresses large time units — e.g., 2 million seconds — into interpretable components such as 23 days, 3 hours, 33 minutes, and 20 seconds.
>
> **[W2]** We will include quantitative analysis of the additional computations. As shown in Figure 2C, after passing through the Transformer backbone, two stages of time generation and time-conditioned multi-scale forecasting occur. The backbone, which accounts for most computations, is used only once, while the remaining operations occur in relatively lightweight heads. This design allows efficient multi-scale forecasting without a significant computational burden.
>
> **[W3]** Thank you for your comment. We acknowledge that we did not provide an adequate explanation of the test loss. The test loss is calculated as a weighted sum of losses across seven downstream tasks, where weights account for the number of relevant samples (e.g., some samples may have IHM labels but not Vasopressor administration labels). This weighted sum can be interpreted as an overall downstream score aggregated across all seven tasks. We recognize this insufficient explanation and will revise it. Additionally, we will include the performance of each downstream task in the appendix.
>
> **[Q1]** A single event can be represented as a Time-Feature-Value triplet. When the Feature and Value are represented as separate tokens (e.g., Figure 1; SBP token, 120mmHg token), this is non-sharing. When combined into a single token (SBP-120mmHg), this is value-sharing. Value-sharing increases the number of tokens required, which can lead to sparse representations for rare events and limit learning. However, combining two tokens into one allows more information to fit within a fixed maximum token length. Since our dataset mainly contains frequently measured vitals and labs, we suspect that value-sharing was not favored in training.
>
> **[Q2]** We performed density estimation for density-based binning using the training set. If the training set is sufficiently large and diverse in terms of age and ethnicity, the risk of out-of-distribution issues should be limited. However, whether density-based binning is more robust than quantile binning under OOD conditions requires further analysis. One potential approach to mitigate distribution shifts is to perform binning separately for major population subgroups (e.g., binning blood hemoglobin levels separately for male and female patients).
>
> **[Q3]** This is a valid point. To control for performance variation due to model size, we unified the backbone across experiments (many baselines use Transformer backbones of different sizes but vary only in loss functions or event representations). Following your suggestion, evaluating performance with smaller backbones could be an interesting extension. Regarding ETHOS, they trained on MIMIC-4 with additional disease codes, labs, and medication data, which differs from our setting. Reproducing ETHOS and comparing under identical conditions would require ~300 A100 hours, which was infeasible for us.
>
> Once again, we sincerely thank you for taking the time to carefully review our manuscript and for your valuable comments.
>
> **Sincerely,**
> *The Authors*

---

### Official Review · Reviewer_m2zP · 2025-11-01

**Soundness:** 3
**Presentation:** 3
**Contribution:** 2
**Rating:** 4
**Confidence:** 4

**Summary:**

This paper introduces a pretraining approach tailored to the special structure of Electronic Health Records (EHRs). The authors propose three main contributions: (1) Pathology-Focused Binning — a density-weighted percentile binning for numerical values that preserves resolution in clinically important (low-density) ranges; (2) Dual-Calendar RoPE — a split rotary positional embedding that jointly encodes relative token order and multi-scale calendrical time (minute, hour, day, etc.); and (3) Time-Conditioned Foreseeing (TCF) — a new dual-objective pretraining head that (i) predicts the next event time in a multi-scale calendrical decomposition and (ii) conditions generation on arbitrary future timestamps to “foresee” events at those times, enabling genuine temporal generative modeling. The method is trained and evaluated on MIMIC-III; it outperforms a broad set of baselines on seven downstream tasks and shows qualitatively and (to some extent) quantitatively superior temporally consistent EHR generation.

**Strengths:**

- The design of the model architecture are well-motivated, including pathology-based binning and temporal positional encoding.

- Strong empirical gains on standard clinical benchmarks. The model consistently outperforms multiple baselines on 8 downstream classification tasks.

- The analysis on generative results are illustrative. The paper includes human and LLM-based evaluation and qualitatively compares the generated results of the model.

**Weaknesses:**

- All experiments use MIMIC-III only. While MIMIC-III is standard, it's quite small and unique. Claims about generality of calendrical modelling and generative realism would be stronger if validated on at least one additional dataset (e.g., eICU, other hospital systems). This limits conclusions about generalization to different EHR systems.

- It is unclear how test loss is defined. Not sure the comparison is fair as the objectives of baselines are different.

- The generative prediction have limited real-world application. The generated timestamps and vital features are random, which can not help clinical decision making. A feasible improvement can be generating prediction for all features at constant temporal gaps, and conduct quantitative analysis comparing to ground truth.

- Figure 6 does not have scale of y-axis.

- The discussion on the failure is insufficient. It would be better to show some failure cases of the model.

**Questions:**

See weaknesses.

---

> ### Author Response · Authors · 2025-12-02
>
> We sincerely thank the reviewer for taking the time to thoroughly read our manuscript and for offering insightful suggestions that meaningfully improve the quality of our work.
>
> We are also grateful for your positive evaluation of our work — summarizing that the model architecture is well-motivated, the empirical performance is strong across multiple clinical benchmarks, and the generative analysis is informative and well-presented — and we appreciate your recognition of these contributions.
>
> We highly value your comments, and as you suggested, **we will expand the training dataset to MIMIC-IV and further explore additional quantitative evaluation strategies to assess the unique generative properties of our method. We will also provide more detailed failure case analyses**.
>
> We believe that revising the manuscript based on your feedback — as well as comments from the other reviewers — will substantially strengthen the paper and enhance its contribution.
>
> Regarding **[W2]**, we agree that our reference to the *test loss* lacked sufficient explanation and requires clarification. During fine-tuning, we optimized the weighted sum of losses across seven downstream tasks, where weights were adjusted based on sample availability (e.g., some samples include IHM labels but lack Vasopressor-related labels). Each task-specific loss was also weighted based on the distribution of positive and negative samples in the training set. Therefore, the reported *test loss* can be interpreted as a weighted overall downstream score aggregated across all seven tasks. We recognize that our explanation was insufficient and will revise this section accordingly.
>
> For **[W3]**, regarding the necessity of the generative component in real-world applications: the generated timestamps are not random — they are learned signals representing predictions of *when* an event will occur. In addition, our model provides a framework for time-conditioned generation. That is, the model can generate what events will occur at specific time points and values such as SBP levels, which can provide clinically meaningful insights. Fixed-interval forecasting (e.g., predicting at constant future gaps) may be suboptimal because clinically relevant temporal resolution varies depending on the situation — from minutes to months. As stated in Line 138, *“In contrast, TCF explicitly models long-range temporal information, thereby capturing how real-world clinical practice unfolds over time.”* This design prevents short-horizon bias and enables long-term temporal reasoning.
>
> Once again, we sincerely appreciate your thoughtful feedback and contribution to improving our manuscript.
>
> **Sincerely,**
> *The Authors*

---

### Official Review · Reviewer_uCLx · 2025-11-01

**Soundness:** 3
**Presentation:** 1
**Contribution:** 3
**Rating:** 2
**Confidence:** 4

**Summary:**

This paper addresses the construction of an EHR foundation model. The authors propose Pathology-Focused binning, which enables density-adjusted binning of numerical observation values, Dual-Calendar RoPE, which incorporates calendrical time in addition to ordinary positional information to capture calendrical periodicity and event concurrency, and Time Conditioned Foresee Objective, which learns when an event will occur. The proposed method was evaluated on tasks commonly used in EHR model evaluation and consistently outperformed baselines.

**Strengths:**

-- Addressing density-adjusted binning, incorporation of calendrical time, and learning when an event will occur is quite a significant problem, and its importance is well-justified in the paper.

-- Experimental results on multiple tasks demonstrated the effectiveness of the proposed method.

**Weaknesses:**

-- The proposed method may have novel points and have some practical impact, but the clarity issues are too severe to understand the method:
* In l.134, l.147, and l.183, E, F, and T are not defined.
* The paragraph "Pathology-Focused Binning" is hard to follow, where a lot of variables and concepts are not defined, such as X, x, h, sigma, and raw count. rho() is used for both functions of x and v, which is also confusing.
* The paragraph "Dual-Calendar Rotary Position Embedding" also contains confusing notations, such as the variable x may be used in a different meaning from the paragraph "Pathology-Focused Binning". What kind of operation is "||"? s and t should be defined. If RoPE() is used as a function, it should be defined.
* l.272 first mentions "the transformer backbone", which looks important, but it does not appear and is not defined before.
* The paragraphs from l.280 are hard to follow since there are many undefined concepts and variables, and equations are not written formally.
* Appendix may contain additional details, but there is an inconsistency of notations from the main text, which makes it difficult to get a clue to understand the proposed method. Also, the main text should be self-contained.
* In Figure 5, what do the bars mean?
* It is better to put the tables and figures on top.

**Questions:**

Nothing.

---

> ### Author Response · Authors · 2025-12-02
>
> We thank the reviewer for taking the time to read our manuscript and for providing thoughtful suggestions that significantly improve the quality of our work.
>
> We also truly appreciate your positive assessment that our paper addresses density-adjusted binning, incorporation of calendrical time, and learning when an event will occur, and that you consider this problem important and well-justified in the manuscript.
>
> We acknowledge your concerns regarding the clarity and presentation of the paper, and we apologize for not conveying our ideas as clearly as we intended.
>
> Following your comments—as well as those from the other reviewers—we will strengthen the Related Work section, further clarify the methodology, and incorporate additional training datasets to enrich the paper.
>
> Once again, we sincerely appreciate your valuable contribution.
>
> Sincerely,
> The Authors

---

### Meta-Review · Area_Chair_gyRq · 2026-01-09

**Summary:**

This paper proposes a time-conditioned generative pretraining framework for EHR data, combining pathology-focused binning, a dual-calendar RoPE time encoding, and a time-conditioned “foreseeing” head for multi-horizon prediction. The problem is important and the empirical section is substantial: the authors re-implement a number of recent EHR foundation baselines under a shared backbone, report improvements on several downstream tasks, and include ablations suggesting that each of the three components contributes. There are also some qualitative analyses of generative behavior. As a piece of applied work within the clinical ML space, the paper is competent and clearly motivated by real modeling challenges in EHRs.

However, I do not think the contribution meets the bar for a top ML conference in its current form. The core ideas are largely incremental engineering refinements of existing concepts—density-aware binning, multi-scale time encodings, and a time-aware output head—rather than a clearly articulated new modeling principle, yet the paper’s framing occasionally overstates its novelty and generative claims. The evaluation is confined to a single dataset (MIMIC-III), which is a narrow basis for “foundation model” or “universal temporal” statements, and the generative side is only weakly validated, with limited quantitative checks of trajectory realism. Several reviewers also noted that the method section is dense and difficult to follow, with notation and exposition that could be significantly streamlined. Overall, this looks more like a solid domain-specific engineering paper that would benefit from clearer writing, more precise positioning, and broader, more rigorous evaluation before it is ready for an ICLR-level venue.

**Reviewer Concerns:**

Reviewer uCLx.
The rebuttal clarified several of uCLx’s concrete issues: the authors explained that their Dual-Calendar RoPE is invariant to year shifts and does not rely on absolute calendar year in a way that would obviously cause temporal overfitting, and they acknowledged that the method description was dense and promised to improve notation and add more formal details and diagrams in the appendix. However, the core concerns remain: experiments are still confined to MIMIC-III (despite the availability of larger datasets such as MIMIC-IV), and the paper’s technical exposition is still relatively heavy and hard to follow. The question of whether the model’s calendrical design truly generalizes beyond the specific dataset and preprocessing choices is not convincingly resolved.

Reviewer m2zP.
For m2zP, the rebuttal addressed several specific points: it clarified how the reported “test loss” is defined as a weighted aggregate over downstream tasks, elaborated on why the time-conditioned foreseeing head is not just cosmetic but intended to provide clinically meaningful temporal signals, and committed to adding more quantitative generative metrics and failure-case analyses. Nonetheless, important issues remain outstanding—most notably that all results are still on a single dataset (limiting claims about generalization of calendrical modeling and generative realism), and that the generative evaluation remains mostly qualitative or indirect for a work that makes strong temporal-generative claims.

Reviewer V1Wk.
The authors responded to V1Wk’s questions about value-sharing, robustness of density-based binning to out-of-distribution values, and model-size sensitivity with additional explanation and some discussion of how the architecture behaves under those scenarios. They also acknowledged that time-conditioned foreseeing and multi-scale prediction add compute and promised clearer ablations and more detailed efficiency reporting. However, the rebuttal does not fully address the reviewer’s broader concerns that the ablation study is limited, that some clinically important metrics are under-explored, and that the gains are shown only in a single medium-scale setting; the depth of empirical analysis and the discussion of trade-offs remain weaker than one would like for a “foundation model” paper.

Reviewer 7R5L.
The rebuttal engages seriously with 7R5L’s comments: the authors agree that generative realism needs quantitative metrics and say they will incorporate model-based checks similar to prior work; they commit to extending experiments to MIMIC-IV; they promise clearer, more step-by-step exposition of the TCF mechanism; and they acknowledge the need to discuss computational cost, fairness, and limitations more explicitly. That said, these are mostly promises for revision rather than evidence in the current submission, so the main weaknesses identified by 7R5L—single-dataset evaluation, lack of quantitative generative metrics, no fairness/subgroup analysis, and limited discussion of failure modes—remain substantively unresolved at review time.

Reviewer FPVU.
The authors respond in detail to FPVU’s extensive critique: they clarify how temporal features are encoded to counter the “absolute year” concern, acknowledge and propose to correct several inaccuracies in the related-work table, and explain their choice of objectives and evaluation strategy. They also indicate they will add stronger baselines and more robust quantitative checks for temporal generation. Even so, FPVU’s fundamental concerns—use of only MIMIC-III for a “foundation”-style claim, absence of simple but strong tabular baselines (e.g., gradient-boosted trees), weak quantitative generative evaluation relative to the strength of the claims, and an overstated novelty narrative given existing temporal EHR models—are not fully addressed in the present experiments and framing.

**Reviewer Scores:**

Reviewer uCLx. This reviewer’s main issues are clarity and single-dataset scope. The rebuttal promises clearer exposition and more formal details but does not change the actual experiments or data coverage. I would expect uCLx to keep essentially the same (borderline-negative) score, rather than moving up.

Reviewer m2zP. m2zP’s concerns are about the narrow evaluation (only MIMIC-III) and limited generative metrics. The rebuttal clarifies definitions and intentions but mostly offers future work rather than new evidence. I would expect m2zP to keep a similar, “mixed but cautious” score, not substantially higher.

Reviewer V1Wk. This reviewer is relatively positive about motivation and empirical gains but wants stronger ablations and broader evaluation. The rebuttal gives more explanation but does not fully satisfy those requests. My guess is that V1Wk would maintain their mildly positive score, not upgrade to a strong accept.

Reviewer 7R5L. 7R5L appreciates the idea but explicitly asks for quantitative generative metrics, additional datasets, and more discussion of fairness and failure modes. The rebuttal mainly commits to adding these later, so their core reservations remain. I expect 7R5L would keep their original “weak accept / borderline” type score.

Reviewer FPVU. FPVU is the most critical and high-confidence reviewer, with substantive concerns about novelty, related work, missing baselines, and overclaiming. The rebuttal corrects some details and promises additional experiments, but does not fundamentally change the picture. I would expect FPVU to leave their (negative) score unchanged.

---

### Decision · Program_Chairs · 2026-01-26

Reject